# Optical tissue clearing and machine learning can precisely characterize extravasation and blood vessel architecture in brain tumors

Serhii Kostrikov [1], Kasper B. Johnsen[1], Thomas H. Braunstein[2], Johann M. Gudbergsson[1,3], Frederikke P. Fliedner[4,5], Elisabeth A. A. Obara[6,7], Petra Hamerlik[6], Anders E. Hansen[1], Andreas Kjaer [4,5], Casper Hempel [1✉] & Thomas L. Andresen [1✉]

Precise methods for quantifying drug accumulation in brain tissue are currently very limited, challenging the development of new therapeutics for brain disorders. Transcardial perfusion is instrumental for removing the intravascular fraction of an injected compound, thereby allowing for ex vivo assessment of extravasation into the brain. However, pathological remodeling of tissue microenvironment can affect the efficiency of transcardial perfusion, which has been largely overlooked. We show that, in contrast to healthy vasculature, transcardial perfusion cannot remove an injected compound from the tumor vasculature to a sufficient extent leading to considerable overestimation of compound extravasation. We demonstrate that 3D deep imaging of optically cleared tumor samples overcomes this limitation. We developed two machine learning-based semi-automated image analysis workflows, which provide detailed quantitative characterization of compound extravasation patterns as well as tumor angioarchitecture in large three-dimensional datasets from optically cleared samples. This methodology provides a precise and comprehensive analysis of extravasation in brain tumors and allows for correlation of extravasation patterns with specific features of the heterogeneous brain tumor vasculature.

[1] Section for Biotherapeutic Engineering and Drug Targeting, Department of Health Technology, Technical University of Denmark, Lyngby, Denmark. [2] Core Facility for Integrated Microscopy, Department of Biomedical Sciences, University of Copenhagen, Copenhagen, Denmark. [3] Laboratory for Neurobiology, Department of Health Science and Technology, Aalborg University, Aalborg, Denmark. [4] Department of Clinical Physiology, Nuclear Medicine & PET and Cluster for Molecular Imaging, Department of Biomedical Sciences, Rigshospitalet and University of Copenhagen, Copenhagen, Denmark. [5] Department of Biomedical Sciences, Rigshospitalet and University of Copenhagen, Copenhagen, Denmark. [6] Brain Tumor Biology, Danish Cancer Society Research Center, Copenhagen, Denmark. [7] Present address: Department of Clinical Biochemistry, Bispebjerg and Frederiksberg Hospital, University of Copenhagen, Bispebjerg, Denmark. ✉email: cash@dtu.dk; tlan@dtu.dk

The blood–brain barrier (BBB) is the most impermeable and selective barrier of the human body. This makes assessment of drug accumulation in the brain particularly challenging, since only low drug concentration can usually be expected in the brain parenchyma[1]. Transcardial perfusion is a technique widely used to clear out the intravascular fraction of the injected compound prior to harvesting tissue samples[2]. This theoretically allows for reducing the contribution from the circulating drug present in the vasculature in bulk measurements, hereby enabling a quantification of extravasated drug exclusively. Different cerebral pathologies are characterized by a varying degree of vasculature and microenvironment remodeling, which may result in an impairment of the regional blood circulation[3,4]. Such remodeling is particularly pronounced in brain tumors like glioblastoma (GBM)[5]. Nevertheless, its impact on the validity of drug accumulation measurements has gone by unrecognized.

GBM is the most malignant type of brain tumor, and, despite intensive research, it's curative treatment remains a challenge[6]. The current first-line GBM treatment strategy, consisting of MRI-guided radical resection with subsequent chemo- and radiotherapy, provides a median survival time of ~15 months from the time of diagnosis[7,8]. Due to the high invasiveness of GBM cells and their ability to migrate in brain parenchyma, complete tumor resection is impossible[9,10]. Furthermore, high cellular heterogeneity provides a solid foundation for tumor multidrug resistance, which, coupled with active angiogenesis, results in fast tumor regrowth despite intensive therapeutic pressure[9,11]. The heterogeneity of GBM blood–tumor barrier (BTB) both with regard to permeability and surface marker profile coupled with inherent cellular plasticity also complicate the development of new treatment strategies[12]. Another factor complicating therapeutic intervention is the intermittent blood circulation in the tumor vasculature, which makes only a part of tumor vessels accessible for the drug at a given time point[5,13].

In order to understand the performance of a drug in such an abnormal and heterogeneous environment, precise methods for evaluating compound extravasation and its spatial distribution are needed[11,14]. Nevertheless, numerous techniques used for this purpose are rather limited in many ways. For example, inductive coupled plasma mass spectrometry (ICP-MS) or high-performance liquid chromatography (HPLC) provide a precise quantitative measure of compound accumulation only at a macroscopic level[15,16], i.e., no spatial details are obtained. Microscopy-based techniques, on the other hand, have lower quantitative precision and throughput, but provide insights into spatial distribution of the compound and its correlation with features of the tumor microenvironment[17,18]. The drawback of the conventional microscopy, however, is that achieving a broad spatial coverage of different GBM regions is challenging due to the need of tissue sectioning. Another important point of consideration is the fact that the precision of multiple methods used for extravasation assessment depends on successful removal of the intravascular fraction of the injected compound by transcardial perfusion[2,19–21].

In the present study, we investigated the phenomenon of transcardial perfusion deficiency in brain tumor vessels, and show that the current methodology of compound extravasation analyses in brain tumors have considerable limitations. As a solution to this, we show the potential of 3D deep tissue imaging for such analysis, and describe machine learning-based image analysis workflows that enable reliable quantification and spatial mapping of compound extravasation, as well as the ability to correlate compound extravasation to the regional characteristics of the GBM angioarchitecture.

## Results

**Transcardial perfusion is inefficient in removing injected compounds from GBM vessels**. For the initial examination of transcardial perfusion efficiency in GBM vasculature, we employed transmission electron microscopy of perfusion-fixed samples of GBM from two orthotropic xenograft models, originating from primary patient-derived GBM cells. The examination revealed tumor vessels filled with erythrocytes (Fig. 1a–c), suggesting inefficient perfusion in the GBM vasculature. This was further corroborated by frequent narrowing of GBM vessel lumens (Fig. 1d–f), which hinders the flow of blood or perfusate. Of note, vessels from the contralateral tumor-free hemisphere showed no signs of insufficient perfusion or limited vascular passability (Fig. 1g–i).

To further study transcardial perfusion deficiency in more depth and to quantify the extent of the problem, we intravenously (IV) injected tetramethylrhodamine (TRITC)-labeled dextran (hydrodynamic radius ~27 nm, ref. [22]) as a drug model, and Alexa Fluor (AF) 647-labeled wheat germ agglutinin (WGA) lectin for tracing the perfusate path (Fig. 2a–d). In order to determine whether the inefficiency of perfusion was a result of insufficient perfusate volume or blood coagulation, three different perfusion protocols were employed (#1 using 20 ml of perfusate, #2 using 55 ml of perfusate, and #3 using 55 ml of perfusate with heparin), based on widely used approaches[23–25]. Images of brains from tumor-bearing animals, which underwent this experimental procedure, showed almost completely homogeneous AF647-lectin mono-labeling of vessels in the tumor-free hemisphere, suggesting full degree of transcardial perfusion in this part of the brain (Fig. 2e–g). In contrast, a heterogeneous pattern of labeling was observed in the GBM vasculature (Fig. 2e–j), illustrating that the IV-injected TRITC-dextran remained in a large part of GBM vasculature, which was not replaced by the lectin-containing perfusate during transcardial perfusion. It was also evident that most of the vessels were either labeled with AF647-lectin or TRITC-dextran only, and that these mono-labeled segments were separated by double-labeled transition zones (Fig. 2h–j). This indicated that heterogeneity in fluorescence signals did not arise from heterogeneous glycocalyx composition in tumor vessels, which could also result in an incomplete lectin binding, but instead arose from inefficient perfusion due to physical obstacles. This pattern was consistent for three different GBM xenograft models (G01, G06 (both primary-derived GBM cells), and U87). To further validate this, we performed high-magnification (63×, 1.4 numerical aperture (NA)) confocal imaging of parts of the vascular tree containing all three types of segments (Fig. 3a–c). These images confirmed continuity between mono-labeled vascular segments and double-labeled transition zones located in between. Collagen IV staining confirmed that the signal from both fluorophores was coming from within the tumor vasculature (Fig. 3d, e). In support of this, analysis of TRITC-dextran intensity showed significantly lower intensity levels in the transition zones compared to mono-labeled (non-perfused) vessels (Fig. 3g, h). This confirmed the assumption of double-labeled segments being partly perfused vessels. To quantify the part of non-perfused and underperfused vasculature in GBM tumors, we employed a trainable machine learning algorithm enabling the segmentation of different types of vasculature, with regard to the degree of transcardial perfusion. Six regions of interest (ROIs) for each tumor and contralateral hemisphere representing a volume of $5.97 \pm 2.49$ mm$^3$ (~10% of tumor volume) for the tumor and $7.12 \pm 1.36$ mm$^3$ for the contralateral hemisphere were segmented by the trained model (Fig. 3f, i–l). The total area covered by each vascular class was quantified and analyzed. The analysis showed that despite large variations between individual brain tumors and

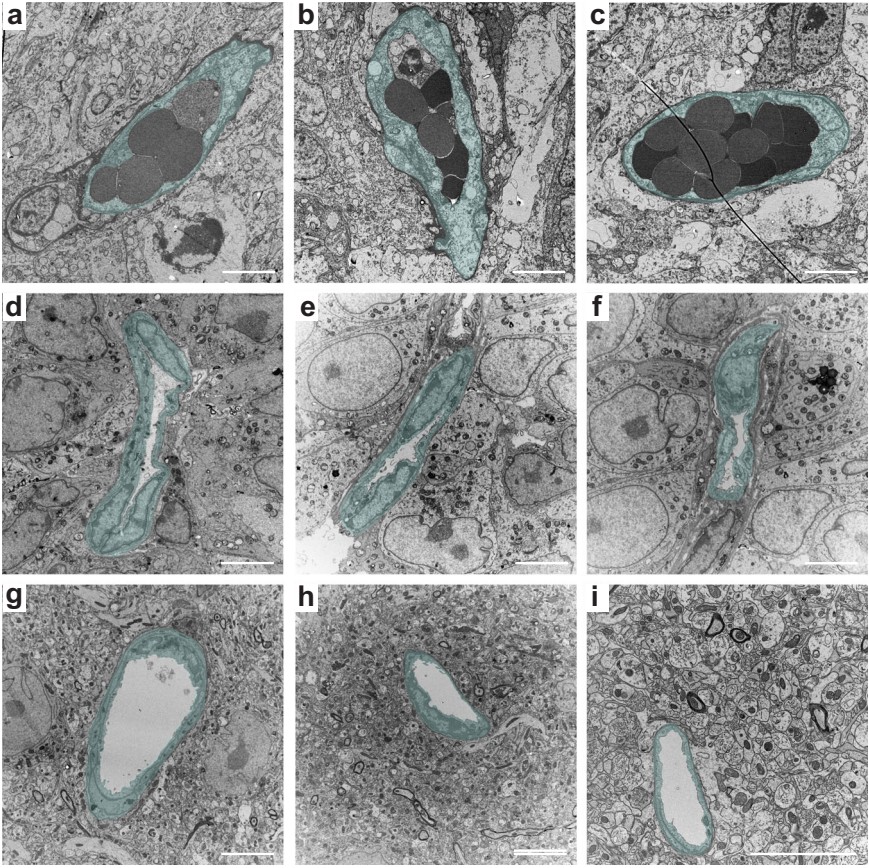

**Fig. 1 Ultrastructural examination of GBM vasculature suggesting incomplete transcardial perfusion.** Transmission electron microscopy imaging of erythrocytes remaining in GBM vessel lumens after transcardial perfusion (**a–c**), compressed vessel lumens in GBM (**d–f**), and the lumens of normal brain capillaries from the contralateral hemisphere (**g–i**). Cells composing vessel walls are highlighted with semitransparent cyan. Scale bars, 5 µm.

the three different GBM cell lines, >50% of functional tumor vasculature was not perfused properly with similar representation of underperfused and non-perfused segments of the vascular tree (Fig. 3m–p and Supplementary Fig. 1). At the same time, vessels of the contralateral hemisphere showed a high degree of perfusion. Here, the percentage of perfused vessels was always >90%, whereas non-perfused vessels never exceeded 4% of the total vascular area. Analysis of the proportions of different vascular classes in tumors versus contralateral hemispheres showed statistically significant differences in all cases, while no significant differences were found when comparing the degree of perfusion in three different GBM models, as well as between the three different perfusion protocols employed (Fig. 3m–p and Supplementary Fig. 1). Importantly, the brain tumor volume was not significantly different between the different experimental situations that were studied (Supplementary Fig. 2d, e). Control experiments in animals not receiving dextran injection, but with subsequent immunohistochemical staining of collagen IV also showed only partial labeling of tumor vessels by the lectins (Supplementary Fig. 3). While the tumor area was characterized by the presence of both double-labeled (lectin + collagen IV) and mono-labeled (only collagen IV) vessels, almost all vessels in the contralateral tumor-free hemisphere appeared as double-labeled. This indicated that dextran injection prior to the transcardial perfusion was unlikely to cause insufficient perfusion of tumor vasculature.

To investigate whether the limited degree of transcardial perfusion in tumor vasculature was a unique phenomenon for GBM, we performed the same experiments in animals bearing flank tumors arising from two different colon cancer cell lines (MC38 and CT26). The results showed either an almost complete

absence of perfused lectin-labeled vessels in the tumor with predominantly non-perfused vessels in MC38 model (Supplementary Fig. 4a, b) or a pattern of heterogeneous fluorescence distribution in CT26 tumors (Supplementary Fig. 4d, e), which was comparable to GBM tumors. Sections from the brain tissue taken from the same animals showed complete lectin labeling of cerebral vasculature, illustrating an otherwise efficient transcardial perfusion (Supplementary Fig. 4c, f). This indicated that the inefficiency of transcardial perfusion observed in GBM vessels may not be restricted to GBM tumors underscoring the need for high-resolution methods, when evaluating compound extravasation in tumor tissues in general.

**Tissue sectioning exaggerates compound extravasation, favoring the usage of optical tissue clearing for extravasation detection.** Aiming at making a clear distinction between the intra- and extravascular fractions of the injected compound in GBM, we studied the performance of conventional microscopy of tissue sections. Examination of low-magnification confocal scans of sectioned tumor tissue showed that many spots, comprising a scarce pattern of dextran extravasation, were located closely to the site of vessel segment ruptures due to tissue sectioning (Fig. 4a, b). This observation led us to hypothesize that some extravasation spots may be artifacts arising from sectioning of non-perfused vessels containing fluorescent compound. High-magnification confocal imaging confirmed the association of tissue section surface and extravasation spots, hereby supporting the notion regarding their artificial origin (Fig. 4e, f). To make sure that this association was not a mere coincidence, but a consequence of sectioning-induced vessel damage, we examined the vasculature in

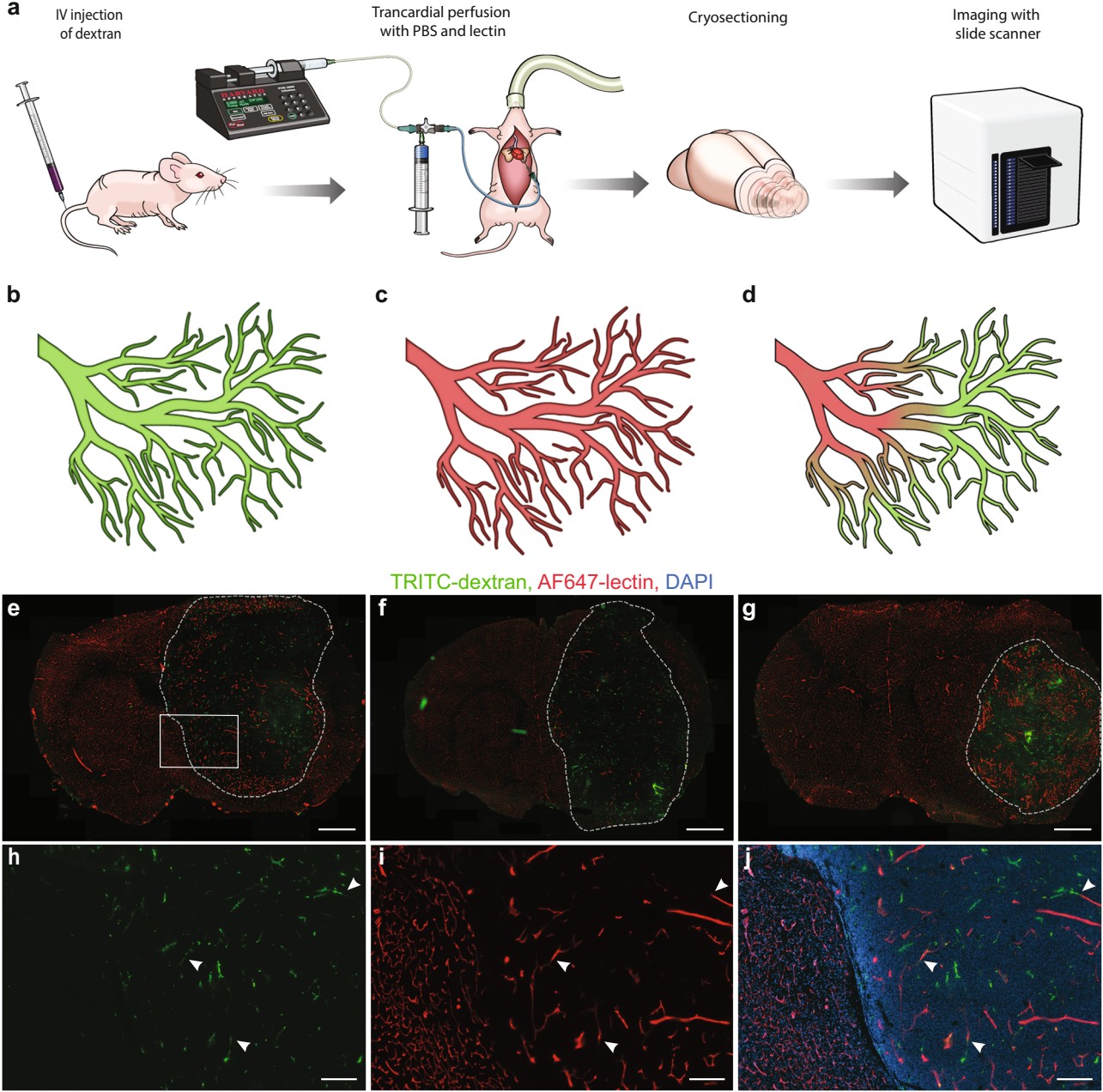

**Fig. 2 Measurement of perfusion degree based on fluorescently labeled probes. a** Schematic representation of the experimental setup for studying the degree of transcardial perfusion. **b** Schematic representation of potential experimental outcomes including full vasculature coverage by TRITC-dextran in a situation with complete inefficiency of transcardial perfusion, **c** full vasculature coverage by AF647-lectin in a situation with successful transcardial perfusion, and **d** heterogeneous vasculature coverage by both AF647-lectin and TRITC-dextran in a situation with incomplete perfusion of varying degrees. **e–g** Representative images of coronal brain sections showing homogeneous AF647-lectin labeling of the vasculature in the tumor-free contralateral hemisphere and heterogeneous distribution of fluorescently labeled probes in the tumor area, illustrating an incomplete perfusion of the tumors arising from G01 (**e**), G06 (**f**), and U87 (**g**) cell lines. Scale bars, 1000 μm. Tumor borders are highlighted with dashed lines. **h–j** High-magnification images of the selected area in **e**. Several examples of transition zones are marked with arrows. Scale bars, 200 μm.

the normal brain harvested without prior transcardial perfusion, where the BBB is uniformly impermeable to 2 mDa dextran. Still, low-magnification confocal images from the normal brain showed dextran extravasation around larger vessels (Fig. 4c, d). High-magnification confocal imaging of the extravasation spots confirmed their location near the edges of tissue sections (Fig. 4g, h) similar to the extravasation spots observed in the tumor (Fig. 4e, f). These observations suggest that compound extravasation studied using microscopy images of conventionally prepared samples

(CPS) may be exaggerated, since a fraction of the extravasation spots arise from sectioning artifacts.

In order to assess the level of overestimation arising from sectioning artifacts, we compared conventional sample preparation and optical tissue clearing, a technique, which achieves a tissue transparency by matching the refractive indices of the tissue sample and the solution, thus allowing large tissue volume imaging and circumventing tissue sectioning (Fig. 4i–k). The appearance of dextran extravasation pattern in optically cleared samples (OCS) was

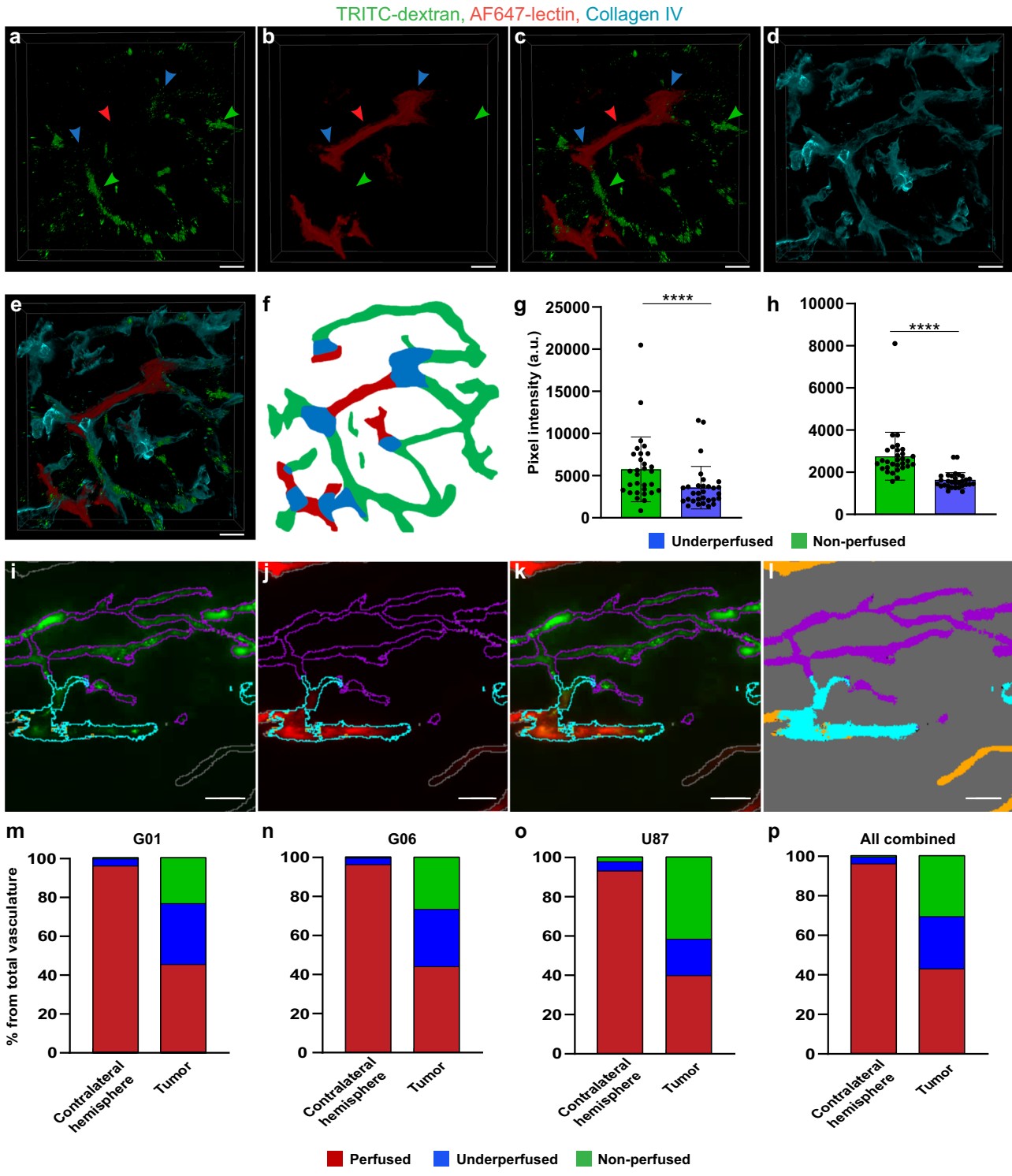

TRITC-dextran, AF647-lectin, Collagen IV

more unambiguous due to preserved structural continuity of tumor vessels (Fig. 4l), ensuring that the registered extravasation spots appeared as a consequence of vascular permeability in the living animal and not due to processing artifacts. Quantification of extravasation spots done by three independent annotators showed a fivefold higher number of spots identified in CPS as when compared to OCS (Fig. 4m). Seemingly, the annotated counts in OCS had a lower variance, suggesting that extravasation in OCS is more unambiguous to quantify (Supplementary Fig. 5). To make sure that extravasation assessment was done in equivalent volumes of tissue, we measured the degree of tissue volume change as a result of sample

preparation for both methods. The dehydration used as a part of clearing process caused ~71% volume decrease, while cryopreservation in sucrose lead to ~12% volume decrease (Supplementary Fig. 6). The sample volume was corrected accordingly. No statistically significant difference in tumor volume was observed between the group of CPS and OCS (Supplementary Fig. 2f).

**A semiautomated machine learning-based image analysis workflow can provide quantification and characterization of extravasation spots in large volumes of GBM tissue.** Image for OCS provide vast amounts of data, so in order to obtain an

**Fig. 3 Analysis of the fluorophores distribution in vessels after IV injection of TRITC-dextran and transcardial perfusion with AF647-lectin. a–c** 3D rendering of high-magnification confocal Z-stacks of GBM vasculature depicting different regions of vascular tree with respect to fluorophores distribution. Green arrows mark non-perfused vessels aligned solely with signal from the remaining TRITC-dextran. Red arrows mark fully perfused vessels as determined by the lack of signal from TRITC-dextran and the presence of signal from AF647-lectin. Blue arrows mark annotation of underperfused transition zones. **d** Vasculature represented by immunostaining of collagen IV in the same region. **e** Overlay of images in **a–d**. Coloring of the vessels segments follows the same logic as for arrows (**a–c**). Scale bars, 10 μm. **f** Diagram illustrating the principles of manual annotation of differently labeled vascular segments for training of the machine learning algorithm. Quantification of the mean pixel intensity of the dextran signal in non-perfused and underperfused vessels in GBM (**g**) ($P < 0.0001$, $n = 30$ mice, Wilcoxon signed-rank test) and in contralateral hemisphere (**h**) ($P < 0.0001$, $n = 30$ mice, Wilcoxon signed-rank test). Data are presented as mean ± SD. Representative images of vessel segmentation performed by the trained model overlaid with original image (**i–k**) and separately (**l**). Segmentation color-coding: magenta—non-perfused vessels, cyan—underperfused vessels, and orange—perfused vessels. Scale bars, 30 μm. **m–p** Quantification of the degree of transcardial perfusion in GBM vessels and vessels from tumor-free contralateral hemisphere. This was studied in animals bearing tumors arising from the G01 (**m**, $n = 10$), G06 (**n**, $n = 10$), and U87 (**o**, $n = 10$). **p** Summary of data from all tumor models ($n = 30$). Proportions of vasculature classes in tumors are significantly different from those in contralateral hemisphere for all groups ($P < 0.0001$, logistic regression after arcsin conversion of percentages), differences in proportions of vasculature classes in the tumors between different GBM models are nonsignificant ($P > 0.3$, logistic regression after arcsin conversion of percentages). More detailed representation of the data in **m–p** with individual values displayed can be found in Supplementary Fig. 1.

efficient semiautomated method for quantification and characterization of compound extravasation in large, cleared GBM samples, we developed a machine learning-based workflow (Fig. 5a–c) consisting of five stages: preprocessing, segmentation, postprocessing, spot recognition, and analysis. In the preprocessing stage, deconvolution of both channels (TRITC-dextran and AF647-lectin) reduced the amount of out-of-focus light and improved the spatial resolution of the images (Fig. 5a, d). For segmentation of extravasation spots, the model was trained on image stacks from different tumor regions and different imaging depths. Some particularly bright intravascular areas of TRITC-dextran gave rise to out-of-focus light in Z-plane due to point spread function elongation (hereafter referred to as PSF-artifacts). Such regions were easily recognized by the human eye, but often wrongly segmented by the initial model. Additional iterative training minimized such misclassification. The trained algorithm produced binary images of extravasation spots, but each spot usually consisted of several "particles" (Fig. 5e), which was sub-optimal for further quantification. Therefore, a postprocessing stage consisting of a series of 2D dilations was implemented to glue "particles" together, while a following Gaussian smoothing brought the dilated spots to their original size and evened up their margins (Fig. 5f). The combination of the abovementioned transformations produced binary masks, which adequately replicated extravasation spots in the original dataset (Fig. 5f, g). The entire workflow is demonstrated in the Supplementary Movie 1. The performance of the workflow was tested in two subsets from different tumors by comparing number of extravasation spots it provided versus numbers counted by the human annotator ("ground truth"). The results showed that the workflow provided bigger numbers in comparison to manual annotation (Fig. 5h). The bigger number mostly arose from the false segmentation of some PSF-artifacts as extravasation spots, which in most cases did not exceed 10,000 μm³ (Fig. 5j, k). Excluding spots smaller than 10,000 μm³ from the datasets, led to results comparable with human annotations (Fig. 5h). These falsely selected spots, despite their big numbers, made only a small contribution to the volume of extravasation (Fig. 5i). With these corrections, the workflow enabled us to measure characteristics, such as volume, surface area, coordinates, etc.

**A semiautomated machine learning-based method allows for vascular tree analysis in large 3D volumes.** To associate extravasation patterns with angioarchitecture, we developed a semiautomated method for comprehensive analysis of the vascular network in GBM. Similarly, to the previously described method, it consisted of five stages: preprocessing, segmentation, postprocessing,

vessel tracing, and analysis. In the preprocessing stage, images underwent blind deconvolution (Fig. 6a) followed by machine learning-based segmentation of vessels (Fig. 6b). To ensure proper segmentation of the vasculature in 3D space, we focused on avoiding inclusion of out-of-focus light, spreading in Z-dimension, into "Object" annotations. This prevented merging of vascular borders of separate, but closely located vascular segments in the Z-dimension. As in case of the extravasation spot analysis, we included Z-stacks from different tumors. In the postprocessing stage, the segmented images underwent slight 3D Gaussian smoothing for initial removal of small isolated regions of false segmentation and smoothing vessel surfaces. To fill the lumens of the vessels segmented as hollow tubes (Fig. 6b and Supplementary Fig. 8a), we developed a flexible workflow, which allowed for more accurate lumen filling and adjusting the size range of the holes to be filled (Supplementary Fig. 7a, b). This was important to avoid the filling of spaces in between small vessels (Supplementary Fig. 7e–j). For further smoothing of the segmented vessels, we applied median filter (Fig. 6c). The hole-filling step was crucial for correct vessel tracing and led to a much more precise recognition of large vessels (Supplementary Fig. 8). This had a considerable effect on the extraction of such angioarchitecture characteristics as volume fraction, branching, length, and diameter of the vessels (Supplementary Fig. 8g–l). However, there were still small parts of the large vessels, which were not traced correctly, but their contribution to the total dataset was negligible. Tracing of the vasculature provided information on multiple parameters of the vascular tree such as vascular volume fraction, mean vessel diameter, mean vessel length, tortuosity, and the number of branching points. The image transformations performed by the workflow are demonstrated in the Supplementary Movie 2. Our workflow enabled a comprehensive analysis of the large vasculature datasets at the resolution of single capillaries (Supplementary Fig. 9). In combination with the extravasation analysis workflow, this can provide a comprehensive quantitative characterization of compound accumulation and its relations with tumor angioarchitecture (Fig. 6e, f).

**Simultaneous application of methods for extravasation and vasculature analysis.** Having developed and refined the methodology, we decided to test its performance using TRITC-dextran and TRITC-albumin as drug models.

For dextran extravasation analysis, two regions with apparently different vasculature from the same tumor were selected (Supplementary Fig. 10a–c) and analyzed (Supplementary Fig. 10d–g). The analysis showed a small degree of extravasation in the form of diffusely distributed small spots with volumes not exceeding 120,000 μm³ with no notable difference between the two regions

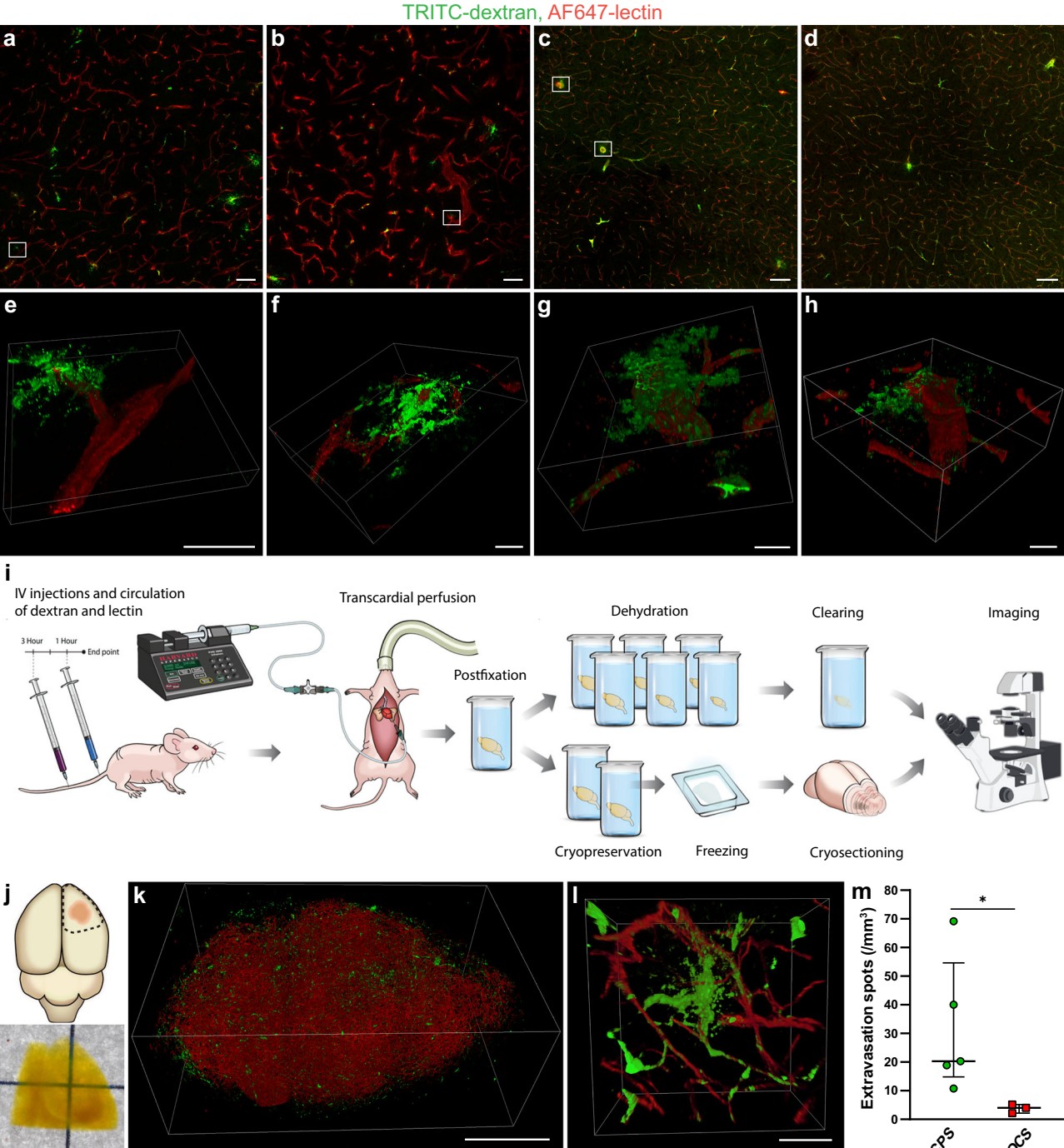

**Fig. 4 Comparison of conventional sample preparation versus optical tissue clearing performance for the assessment of compound extravasation in GBM. a**, **b** Confocal images of TRITC-dextran extravasation in GBM. **c**, **d** Confocal images of TRITC-dextran extravasation in tumor-free brain regions. Scale bars, 100 μm. **e**, **f** 3D rendering of high-magnification confocal Z-stacks of the regions marked in **a** and **b**, illustrating extravasation spots being located in a close proximity to the sectioning-induced vessel disruption near the surface of tissue section. **g**, **h** 3D rendering of high-magnification confocal Z-stacks of the regions marked in **c**, illustrating the same phenomenon in normal brain vessels. Scale bars, 20 μm. **i** Schematic representation of the experimental setup employed to compare the performance of standard microscopy of conventionally prepared samples versus 3D deep microscopy of optically cleared samples for detection and quantification of compound extravasation in GBM. **j** Schematic depiction of the sample excision from the brain (top), and a bright-field image of the sample after clearing procedure (bottom). **k** 3D rendering of a GBM tumor in its full volume. Scale bar, 1000 μm. **l** 3D rendering of a high-magnification confocal Z-stack containing a TRITC-dextran extravasation spot arising from an intact tumor vessel. Scale bar, 50 μm. **m** Quantification of extravasation spots in G01 tumor samples, which were optically cleared ($n = 3$) or prepared conventionally ($n = 5$). Data are presented as median values of the estimates made by three researchers per tumor, bars represent medians with interquartile range ($P = 0.035$, Mann–Whitney test). CPS conventionally prepared samples, OCS optically cleared samples.

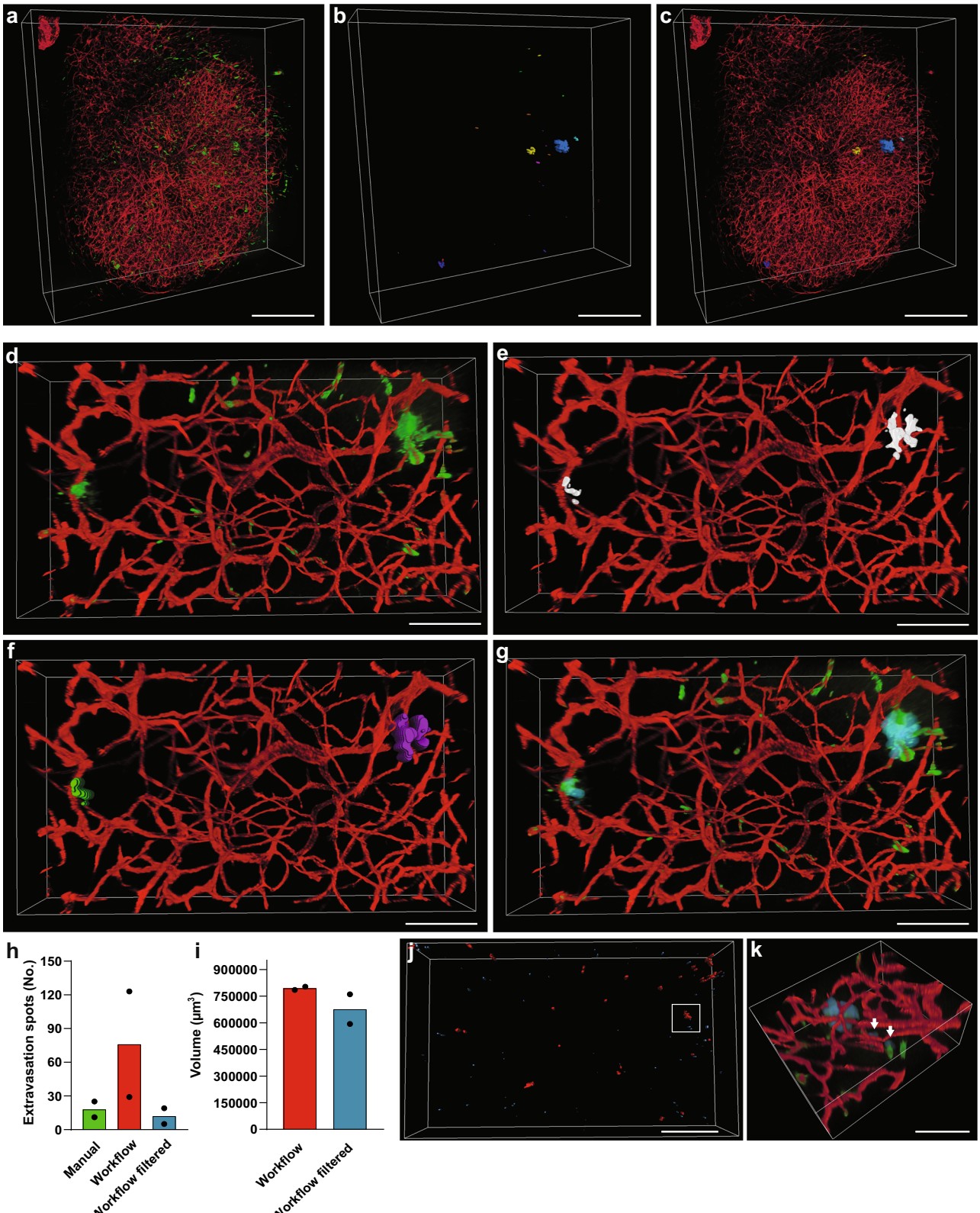

(Supplementary Fig. 10h, i). Vasculature analysis allowed us to detect and quantify differences between the two subsets on such parameters, as volume fraction, mean vessel diameter, and percentage of the vascular volume distribution between vessels with different diameters (Supplementary Fig. 10j, l, m). No apparent differences were seen in mean segment length, number of branching points, and vessels tortuosity (Supplementary Fig. 10k, n, o). Thus,

tumor vessel permeability to TRITC-dextran was likely not linked to any of the specific characteristics of vessel geometry measured in this particular GBM tumor.

We then applied a combination of thresholding-based analysis of TRITC-albumin extravasation and vasculature analysis to inspect relations between GBM angioarchitecture and vessel permeability. Two tumor regions with apparently different

**Fig. 5 A semiautomated machine learning-based workflow for detection and analysis of extravasation spots in 3D deep images. a** 3D rendering of a large data subset of a cleared tumor derived from an animal after IV injections of TRITC-dextran (green) and AF647-lectin (red). **b** Segmented and recognized TRITC-dextran extravasation spots from the dataset in **a** colored independently of any attributes. **c** Overlay of the original dataset with the segmented and recognized TRITC-dextran extravasation spots. Scale bars, 500 μm. **d** 3D rendering of tumor region containing dextran extravasation spots. **e** Extravasation spots in **d** segmented by trained machine learning algorithm. **f** Segmented extravasation spots in **e** after postprocessing steps recognized in Amira as separate "particles". **g** Overlay of the original dataset with the segmented extravasation spots (in semitransparent cyan). Scale bars, 100 μm. **h** Quantification of extravasation spots by human annotator and the developed workflow before and after correction for PSF-artifact contribution. Data are presented as mean with individual dots representing individual samples. **i** Quantification of the total extravasation volume measured before and after correction for PSF-artifact contribution. Data are presented as mean with individual dots representing individual samples. **j** Recognized extravasation spots in one of the quantified samples (red spots > 10,000 $\mu m^3$ and blue spots < 10,000 $\mu m^3$). Scale bar, 500 μm. **k** High-magnification image of the region marked in **j** overlaid with the original data. Segmented extravasation spots are depicted in semitransparent blue. Arrows depict falsely recognized extravasation spots due to PSF-artifacts. Scale bar, 100 μm.

leakage profiles were selected (Fig. 7a–c) and analyzed (Fig. 7d–g). A high extravasation of albumin (Fig. 7h) was accompanied by a number of differences in angioarchitecture. In the more permeable tumor region, mean vessel diameter was increased compared with the less permeable region. This was supported by an increased vascular volume fraction with bigger percentage of total vessel volume being represented by vessels with larger diameter. It was furthermore characterized by an increased branching and vascular volume fraction (Fig. 7i–k, m and Supplementary Fig. 9c). It was also characterized by a slightly increased number of large and midsized segments, and a decreased number of small segments (Supplementary Fig. 9d). Mean branch length and tortuosity were comparable in the two regions (Fig. 7l, n). These data exemplify an association of the vascular permeability degree in GBM with some of the angioarchitecture parameters.

## Discussion

Precise methods for studying compound extravasation in GBM are important for our understanding of any given drug performance, and, hence, its further clinical success. However, certain features of GBM biology pose serious methodological limitations to this. As we show in the present study, deficient transcardial perfusion and the resulting persistence of compounds in the vasculature of GBM, as well as in other types of tumors may substantially impact our interpretations of drug extravasation levels. The methods developed here not only enable a precise measurement of compound extravasation and its spatial distribution despite these limitations, but also allow to investigate its connections with regional pathology of tumor vasculature.

Transcardial perfusion is known to be an indispensable tool for acute drug extravasation experiments. By removing the intravascular fraction of an injected compound, it produces samples with only the extravasated compound fraction for further assessment[2,26,27]. Nevertheless, in our hands the technique fails to remove the injected compound from almost half of the functioning vessels across different GBM mouse models, regardless of perfusion protocol. We assume that such a perfusion deficiency can at least partly be attributed to intermittent perfusion of tumor vasculature, which is the reason for tumor cycling hypoxia[13]. A solid body of evidence from both preclinical and clinical studies suggests that a considerable part of vessels in solid tumors have periods of partial or complete absence of blood flow due to obstruction and collateral shunting of blood[13,28–30]. The kinetics of the process is rather complex, and consists of superimposed cycles with different durations ranging from minutes to hours and even days[28,31–33]. To mimic the situation in acute experiments for compound extravasation studies, TRITC-dextran was allowed to circulate for 3 h, so it was distributed among the vessels with normal perfusion and in intermittently perfused vessels with shorter perfusion cycles. During transcardial perfusion, we assume that many intermittently perfused parts of the

vascular tree containing TRITC-dextran may not be accessible to the perfusate, hereby trapping TRITC-dextran inside.

Even though the phenomenon of intermittent vascular perfusion in GBM has been known for years[34], it has not been considered as a factor causing serious methodological limitations in drug delivery studies, and transcardial perfusion has been widely employed for blood removal in tumor vessels[19,21,35–37]. In our study, we quantified the part of GBM vasculature, which retains a substantial portion of injected compound due to transcardial perfusion deficiency and thus exaggerate compound extravasation. These findings suggest that relying only on quantitative techniques that cannot provide a clear distinction between the extra- and intravascular fractions of the injected compound can lead to erroneous quantification of drug accumulation. Among such techniques are, for example, measurements performed on tissue lysates (e.g., ICP-MS, HPLC, etc.) or low-magnification fluorescent measurements (e.g., small animal fluorescence imaging systems)[15,16,38]. Microscopy, on the other hand, can provide the necessary distinction between compound fractions[17]. Yet, conventional confocal microscopy of sectioned samples will also result in exaggerated picture of compound extravasation due to sectioning artifacts. This could be avoided by designing the experiments such that the time point of analysis would be long enough to have the compound cleared from the systemic circulation. However, some compounds may circulate for up to several days[39], and therefore the tumor samples taken in the end of this period will not reflect acute compound extravasation. To overcome these limitations, we suggest using 3D deep imaging of optically cleared tissue. By keeping the GBM vasculature intact, optical tissue clearing solves the problem of sectioning artifacts due to retained compound, and provides the least distorted results. Among other advantages of the technique is the fact that it provides a comprehensive 3D view on the compound distribution in GBM, which is very important if the latter is heterogeneous, and it requires much less hands-on time for producing an image of a given sample in comparison to imaging of multiple tissue sections. It is also important to note that various tissue clearing methods often use aggressive compounds[40], which can have versatile effects on the tissue, as well as extravasation patterns, and are yet to be comprehensively studied. Such potential effects can be limited by using optical clearing methods with relatively mild tissue treatment like the one described in the present study, on one hand, and, on the other, by ensuring robust chemical fixation of the studied compound and the tissue sample in general.

3D deep imaging of cleared samples produces massive amounts of data, which can be fully utilized only by applying appropriate image analysis methods[40,41]. However, image analysis of such datasets is still rather challenging and highly complex, representing a bottleneck in adopting optical tissue clearing by a broad biomedical community. Segmentation of large 3D datasets has

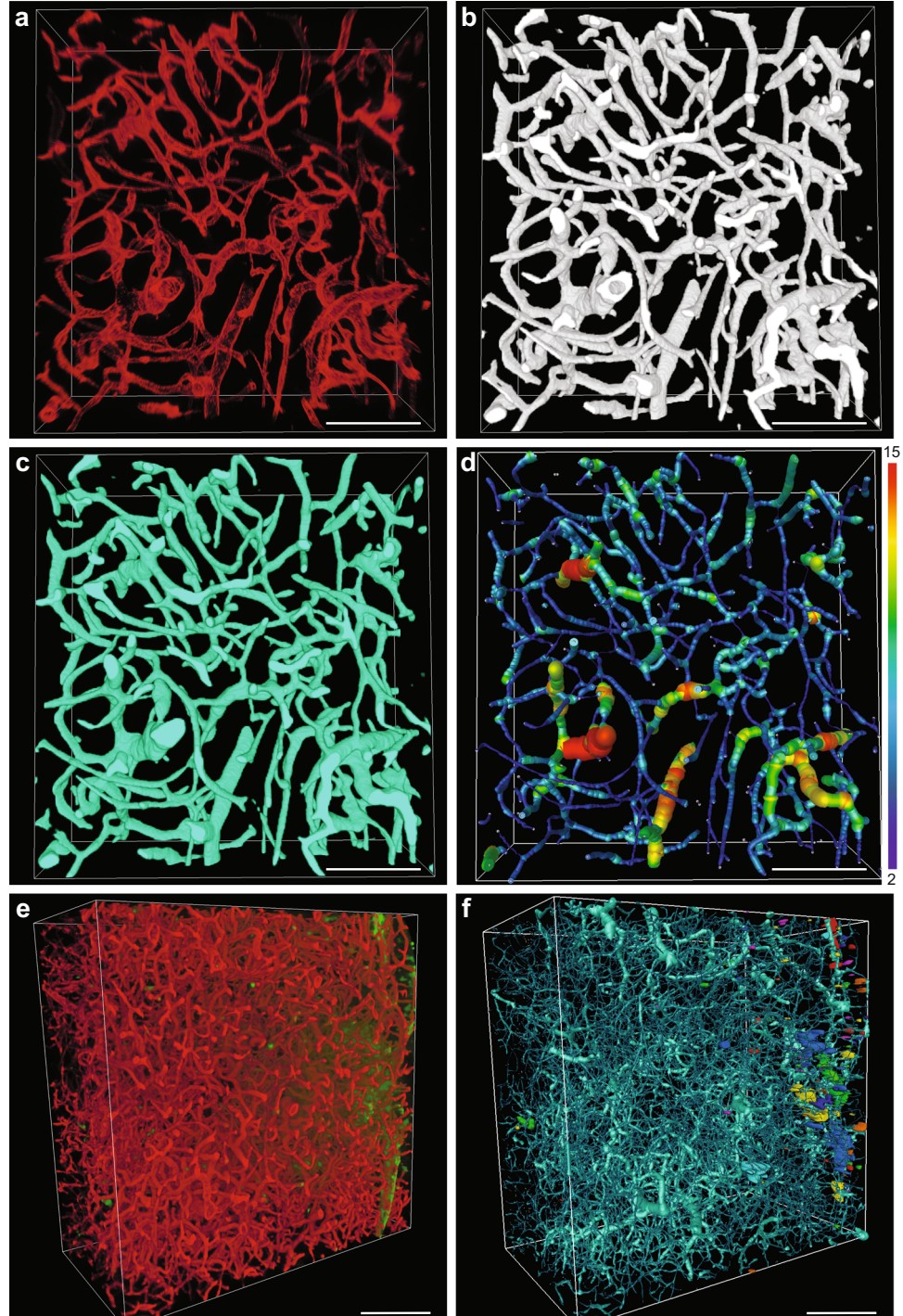

**Fig. 6 A semiautomated machine learning-based workflow for segmentation, tracing, and analysis of GBM vasculature in 3D deep images. a** 3D rendering of AF647-lectin-labeled tumor vasculature. **b** 3D rendering of vessel segmentation performed on the dataset in **a**. **c** 3D rendering of segmented vessels after postprocessing steps. **d** 3D rendering of traced GBM vasculature with color-coding according to the vessel diameter, numbers for minimum and maximum vessel diameter are presented in μm. Scale bars, 100 μm. **e** 3D rendering of AF647-lectin-labeled GBM vasculature (red) with extravasation of TRITC-dextran (green). **f** Dataset from **e** with the extravasation spots recognized (colored independently of any attributes) and the vasculature traced. Scale bars, 200 μm.

always been problematic due to inhomogeneous levels of signal and signal-to-background ratio throughout sample regions[42], making it very difficult to achieve satisfying results with thresholding-based approaches. Therefore, in our workflows, we implement the segmentation method based on deep learning, which allows for pixel classification based on a multitude of

parameters. It is also important to note that while the exemplified segmentation approach takes advantage of deep learning (through employing a convolutional neural network with pre-trained weights for feature extraction, which are used afterwards for pixel classification by random forest of decision trees), it is not as time-consuming as training of a neural network from

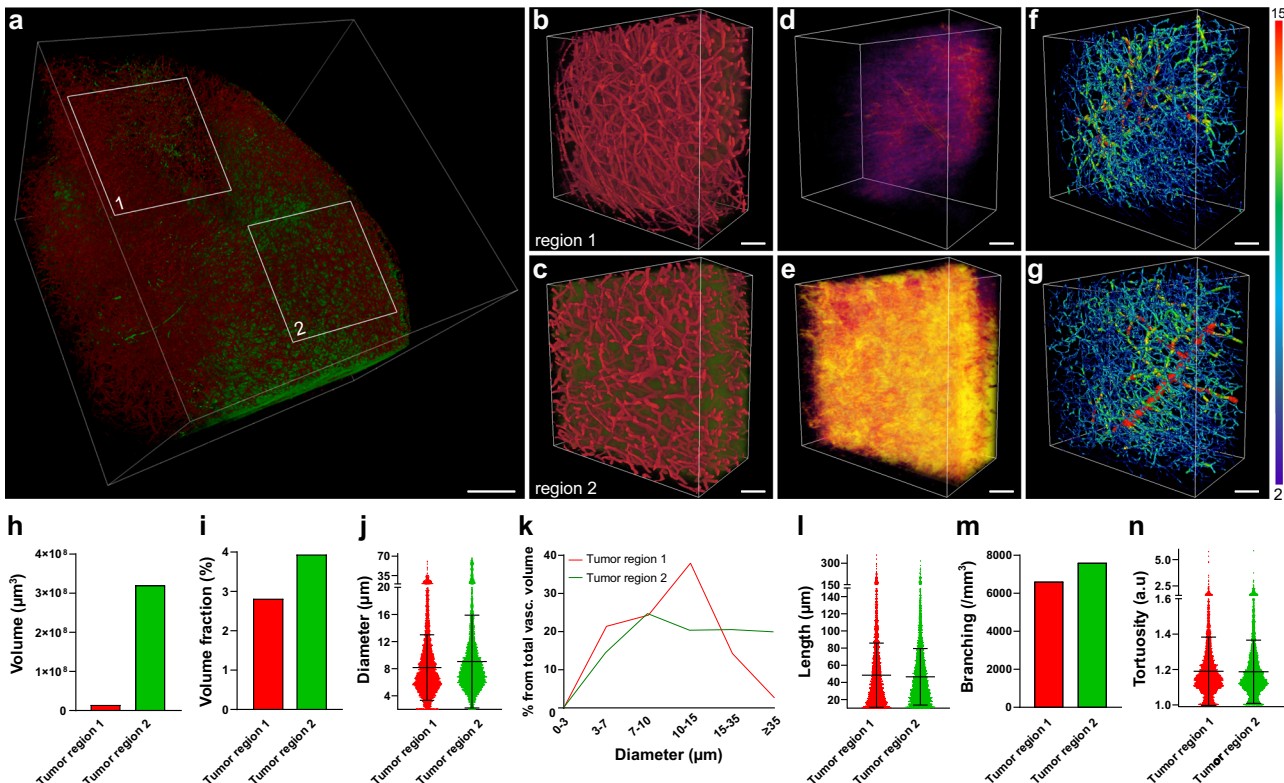

**Fig. 7 Method application example: relationship between GBM vasculature permeability to TRITC-albumin and the angioarchitecture. a** 3D rendering of a large data subset derived from a cleared tumor dissected from an animal after IV injections of TRITC-albumin (green) and AF647-lectin (red). Scale bar, 500 μm. **b, c** 3D rendering of the regions marked in **a** in higher magnification. **d, e** TRITC-albumin extravasation segmented by thresholding after subtraction of the segmented vasculature in the tissue subsets in **b** and **c**, the colormap reflects the density of the pixels in the extravasation mask. **f, g** Traced vasculature from the subsets in **b** and **c** colored according to the vessel diameter, numbers for minimum and maximum vessel diameter are presented in μm. Scale bars, 100 μm. **h** Total volume of extravasation in the tissue subsets. **i** Vascular volume fraction in the tissue subsets. **j** Mean diameter of the vascular segments in tissue subsets. Data are presented as mean ± SD ($n = 4804$ and 5387 individual vessel segments). **k** Histogram representing the volume fraction occupied by vessels of a certain diameter from the total vascular volume fraction in the tissue subsets. **l** Mean length of the vascular segments in the tissue subsets. Data are presented as mean ± SD ($n = 4804$ and 5387 individual vessel segments). **m** Number of branching points per volume in the tissue subsets. **n** Mean tortuosity of the vasculature in the tissue subsets. Data are presented as mean ± SD, ($n = 4804$ and 5387 individual vessel segments).

scratch[43]. This is because the used neural network (VGG-19) is already pretrained and training is required for the classifier only, which needs orders of magnitude less training data compared to a neural network. Another important issue with workflows for the advanced analysis of large 3D datasets is their adaptability. Implementation of such workflows usually require a certain degree of familiarity with one or more programming languages like Python or Matlab[18,42,44]. Therefore, one of our main considerations for designing the workflows presented in this paper was to ensure that they are relatively easy to apply and customize for people, with no or little programming expertise. This was achieved through incorporating the software with developed user interfaces, such as ZEN Intellesis, Fiji, and Amira. Such workflows are in general lacking in the currently available literature.

There can be different approaches to the analysis of compound extravasation in 3D deep images and they depend on the features of extravasation pattern. The first scenario is massive extravasation, when the compound diffusively distributes in a major part of extravascular space like in the case of albumin extravasation in the present study (Fig. 7a–e). In this case, thresholding-based segmentation can be justified and does not cause serious distortion of the extravasation picture. Another scenario is a less pronounced compound extravasation, which is observed in the proximity to most tumor vessels. In this case, the extravasation pattern can be analyzed with respect to the distance it has traveled

from the vascular wall by creating distance maps[45]. Yet another scenario is scarce extravasation with focal compound accumulation in close proximity to the vessels as the one observed for TRITC-dextran in the present study (Fig. 5a–g). Analysis of such patterns requires particularly high precision, which can be obtained by employing machine learning-based image segmentation and proper image postprocessing. Furthermore, due to high intra- and intertumoral heterogeneity in GBM[46], conclusive characterization of a compound's ability to penetrate tumor tissue requires extraction of multiple data points describing the extravasation pattern. Together with general parameters, such as total extravasation volume, the workflow described here allows to extract data regarding spatial differences in compound extravasation at the resolution of a single extravasation spot. At such a level of detail, the method can detect even subtle differences in the performance of drug delivery systems, and thus, provide valuable information for improvement of drug delivery system design. In addition, the method can be utilized for immune cells tracking and analysis of pathological deposits, such as amyloid-β (refs. [47,48]).

Another important aspect of characterizing a drug delivery system's performance is defining the vascular properties and features of the tumor microenvironment in the studied regions. The GBM vasculature is remarkably heterogeneous due to two interconnected factors: the numerous mechanisms of angiogenesis[12,49,50], and the

heterogeneous tumor microenvironment[46]. A growing number of studies suggest that these parameters are, at least in part, reflected by tumor angioarchitecture. Therefore, a detailed analysis of these parameters can provide insights into drug delivery system performance in the different microenvironments in GBM[51]. At the early stages, GBM cells co-opt relatively small vessels[52]. Subsequently, vessels in the vicinity to the GBM cells become larger than average capillaries in the normal brain and more permeable due to tumor cells displacing astrocytic end feet[52,53]. With the rapid growth of the tumor bulk, some regions continue its invasive growth by hijacking normal brain vessels, while other regions provide vascularization by inducing intensive sprouting angiogenesis, a process dependent on vascular endothelial growth factor (VEGF) signaling[54]. Such regions are characterized by high vascular permeability, abundant areas of hypoxia, necrosis, and a pronounced mass effect[20,54]. At the level of the angioarchitecture, these conditions are reflected by increased vasculature volume fraction and vessel diameter[20]. As hypoxia increases, leading to an increase in VEGF signaling, vessels become more permeable, which is reflected by the diameter increase at the level of angioarchitecture[20,55]. It is important to emphasize that clinical data also show significant correlation between vessel diameter, BTB permeability, and tumor grade[56]. Obviously, more correlative studies are needed to establish connections between the numerous characteristics of angioarchitecture and other aspects of tumor biology. Nevertheless, angioarchitecture analysis has a big potential for becoming a very useful technique also in clinical practice. Not only can it improve biopsy phenotyping[57], but it may also advance noninvasive tumor evaluation, especially with further improvement in resolution of medical imaging techniques[58,59].

Analysis of GBM angioarchitecture alongside with extravasation assessment poses a number of challenges and calls for a dedicated approach. While thinning operations, used for skeletonization of 3D networks, require all vessels to be segmented as filled tubes[60], labeling of vessel lumens in GBM is practically impossible in experiments for extravasation analysis. Due to deficiency of transcardial perfusion, the labeling agent has to be introduced by IV injection, and, as our experiments showed, after 1 h of circulation, AF647-lectin reached all GBM vessels accessible for TRITC-dextran during 3 h of circulation. The need for a long circulation time of the vascular labeling agent makes it impossible to employ gelatin-based vessel labeling strategies, which provide staining of vascular lumen[61,62]. Also, due to the BTB leakiness, the vessel labeling agent must have high affinity to the internal vascular surface to ensure that signal coming from vasculature is much stronger than the signal from its extravasated fraction. Such a labeling strategy, however, cannot provide vessel lumen labeling in the large perfused tumor vessels, which are therefore segmented as hollow tubes. Filling of such vessels has to be done at the image processing stage, but the usage of morphological closure tools (dilation/erosion operations) is limited due to high variability of GBM vessel size (Fig. 7j–l and Supplementary Fig. 9)[63]. Furthermore, the usage of an alternative option, such as a standard nonadjustable hole-filling tool, which fills all empty spaces surrounded by signal from four sides in 2D space, can affect the regions with dense capillary networks (Supplementary Fig. 7e, g, i). We addressed the abovementioned challenges by utilizing the high precision of machine learning-based segmentation, which minimizes the need for postprocessing operations and developed adjustable Fiji-based method for vessel lumen filling.

In conclusion, we showed that transcardial perfusion deficiency in tumor vessels has major implications for the assessment of compound extravasation in GBM, and that many of currently used methods would overestimate extravasation. We demonstrated how 3D deep imaging of cleared samples is not subjected to the limitations of vascular underperfusion and can robustly assess drug delivery system performance. We also developed user-friendly complementary image analysis techniques allowing to characterize compound extravasation patterns in large 3D datasets and correlate it with angioarchitecture parameters. We believe that this can substantially improve the assessment of GBM targeting and provide a deeper understanding of the drug distribution in tumor tissue, which is much needed for fighting GBM.

## Methods

**GBM cell lines and culturing.** For the present study, three GBM cell lines of different origins were used: G01, G06, and U87MG. Primary GBM cells (G01 and G06) were obtained from freshly resected tumor tissue obtained from the Department of Neurosurgery at Copenhagen University Hospital in accordance with the guidelines outlined by the Danish Ethical Committee and the Danish Data Protection Agency (H-3-209-136_63114), including informed consent prior to surgery. The G01 and G06 cell lines were passaged subcutaneously in the flanks of NOG mice as outlined in the Danish Welfare Law on Animal Experiments Act no. 1306, protocol: 2012-15-2934-00636. Single cells were isolated from xenografts using the dissociation system (Worthington Biochemical Corporation, #LK003150), according to manufacturer's instructions. Isolated cells were grown in Neurobasal-A medium supplemented with B27 minus vitamin A (Thermo Fisher Scientific, #12587-010), epidermal growth factor (20 ng/ml, R&D systems, #236-EG-01M), fibroblast growth factors (20 ng/ml, R&D systems, #4114-TC-01M), GlutaMAX (Thermo Fisher Scientific, #35050-038), and antibiotics (Thermo Fisher Scientific, #15140122). Both lines were routinely authenticated using ATCC STR profiling. U87MG cells (ATCC and HTB-1) were grown in Dulbecco's Modified Eagle medium (DMEM) with GlutaMAX (Thermo Fisher Scientific, #10566032) supplemented with fetal calf serum (100 mg/ml) and antibiotics (Thermo Fisher Scientific, #15140122). All cell lines were routinely tested for mycoplasma.

**GBM orthotropic xenografts.** Female NMRI-nude mice aged 7–9 weeks (Taconic Biosciences, # NMRINU-F) were inoculated orthotopically with GBM cells grown as neurospheres in Neurobasal-A medium (12349-015, Invitrogen) supplemented with B27 minus vitamin A (Invitrogen, #12587-010), epidermal growth factor (20 ng/ml), fibroblast growth factors (20 ng/ml), GlutaMax (Invitrogen, #35050-038), and antibiotics (Invitrogen, #15140-122). Prior to inoculation, the GBM neurospheres (G01, G06) or GBM cells grown as monolayer (U87MG) were treated with TrypLE (Thermo Fisher Scientific, #12604013) to generate a single cell suspension and resuspended in PBS reaching the concentration of 5000 cells/μl (G01 and G06) and 20,000 cells/μl (U87MG). Animals were anaesthetized with intraperitoneal injection of Hypnorm/Midazolam mixture and placed in the stereotaxic frame. The surgical area was locally anaesthetized by subcutaneous injection of Marcain and rinsed with alcohol. An incision was made in the scalp and a burr hole drilled 1 mm anterior, 1.6 mm laterally from Bregma. The needle of a Hamilton syringe was first positioned at the depth of 2.5 mm from dura and then moved 0.5 mm upward creating a small void preventing the backflow of the cell suspension. A total of 10,000 (G01 and G06) or 200,000 (U87) cells were inoculated into the striatum using a micro pump (World Precision Instruments, UMP3) running at the speed of 1 μl/min. As an additional preventive measure for avoiding backflow of the cells, the needle was removed from the inoculation site 3 min after the end of injection. Tumor progression was monitored with T2-weighted MRI (Bruker, BioSpec 7T, acquisition software: Paravision 360) and a body weight of animals was measured three times a week. The criteria for performing experimental procedures were at least 3 weeks of tumor growth and a tumor volume exceeding 10 mm³. Symptoms such as a body mass drop of >15%, paresis, plegia, or hypoactivity were criteria for endpoint. Volumes of all tumors used for the present study are represented on Supplementary Fig. 2g. Tumor volume was measured with Horos 3.3.5 (Horos Project) on the scans taken within 24 h before experimental procedure. All experimental procedures involving animals were conducted under the approval of institutional boards for ethical use of experimental animals (Copenhagen University and Technical University of Denmark) and Danish Animal Experiments Inspectorate (license #: 2020-15-0201-00482). The experiments were performed in accordance with relevant guidelines and regulations (EU directive 2010/63/EU and relevant local additions to this directive).

**Syngeneic colorectal cancer models.** Two types of tumors models (arising from CT26 and MC38—syngeneic colorectal cancer cell lines) established in 7–10-week-old female BALB/cJRj and C57BL/6JRj mice (Janvier). CT26 cells were cultured in RPMI (Thermo Fisher Scientific, #61870044) with fetal bovine serum (100 mg/ml, Thermo Fisher Scientific, #10500064), GlutaMAX (2 mM), and antibiotics. MC38 cells were cultured in DMEM with fetal bovine serum (100 mg/ml), GlutaMAX (2 mM), and antibiotics (Thermo Fisher Scientific, #15140122). At the day of inoculation, mice were anaesthetized with sevoflurane and 300,000 cells were injected into the flank region. CT26 cells were inoculated in serum-free RPMI and MC38 cells were inoculated in serum-free DMEM. Tumor volumes were calculated as length × width² × 0.5 based on digital caliper measurements.

**Electron microscopy.** Tumor-bearing mice were transcardially perfused with PBS followed by 2.5% glutaraldehyde in 0.1 M sodium cacodylate buffer (Caco; EMS, #11653). After overnight postfixation, samples were taken from the tumor and

contralateral hemisphere using a biopsy needle (diameter: 1 mm). The samples were washed in 0.15 M Caco buffer containing 2 mM calcium chloride three times (first time—overnight at 4 °C, and two times at room temperature (RT) 15 min each) and postfixed in the solution of 2% osmium tetroxide (Sigma-Aldrich, #75632) and 1.5% potassium ferrocyanide (Sigma-Aldrich) in 0.15 M Caco buffer (pH 7.2) for 20 min at RT. After 3 washes (5 min each at RT) in distilled water, samples were immersed in 1% thiocarbohydrazide (Sigma-Aldrich, #88535) solution in distilled water. After 30 min, tissue was washed 3 times with distilled water and left in 2% osmium tetroxide for 1 h. After 3 washes with distilled water, uranyl bloc stain was performed by immersing the samples in 1% uranyl acetate (EMS, #22400) at 4 °C overnight. Then, samples were washed and immersed in the solution of 0.02 M lead nitrate (Sigma-Aldrich, #467790) in 0.03 M sodium aspartate for 30 min at 60 °C. Afterwards, the samples were washed and dehydrated in ethanol solutions 70% (2x), 90% (2x), 96% (2x), 100% (2x + 2x anhydrous ethanol) 7 min each step at RT and immersed in propylene oxide 100% (Sigma-Aldrich, #20401) 2x. The samples were then consecutively moved to 25%, 50% and 75% solutions of epon hard (TAAB, #812 resin T031) in propylene oxide (each immersion step lasting 30 min) and left overnight in 100% epon hard. For polymerization, epon-embedded samples were left at 60 °C for 24 h. Ultrathin sections were cut and imaged using transmission electron Microscopy (Philips, CM100).

### Evaluation of transcardial perfusion degree in tumor vasculature

*Perfusion.* Tumor-bearing mice received IV bolus injections of lysine-fixable TRITC-conjugated 2 mDa dextran solution (50 µg/g, Thermo Fisher Scientific, #D7139). The IV injection flow rate was kept <1 ml/min. After 3 h of TRITC-dextran circulation, mice were anaesthetized with sevoflurane and transcardially perfused. In order to remove the major fraction of erythrocytes, the mouse was first perfused with 5 ml of PBS, followed by the perfusion of 5 ml AF647-conjugated WGA (100 µg/ml, Thermo Fisher Scientific, #W32466) in PBS aimed at marking the intravascular path of the perfusate. The next step of the perfusion procedure varied depending on the perfusion protocol used: in protocol #1—perfusion of 10 ml PBS, in protocol #2—perfusion of 45 ml PBS, and in protocol #3—perfusion of 45 ml PBS with heparin (10 i.u./ml, LEO Pharma). In order to standardize perfusion flow rate, the flow of the perfusate was driven by a syringe pump (Harvard Apparatus, #0-2001) running at a flow rate of 8 ml/min. The flow rate was selected to match the cardiac output of the mouse[64] and was proven effective for perfusion of the normal brain vessels (Figs. 2e–j, 3m–p, and Supplementary Fig. 1). The only interruption of the perfusion was a short break for manually injecting the lectin solution to the perfusion system.

*Tissue processing and imaging.* After perfusion, brains were dissected, immersed into frozen section medium (Leica Biosystems, #3801480), and snap-frozen in liquid nitrogen-cooled isopentane (Sigma-Aldrich, #NC0948209). 60 µm coronal sections were obtained using a cryostat (Leica Biosystems, 1850 UV). In order to cover different tumor regions, six sections were taken from each tumor. First and last sections represented most anterior and posterior tumor parts. The size of the interval for section collection was constant and was calculated based on sagittal MRI scans. Sections were mounted with DAPI-containing mounting media (Abcam, #ab104139) and imaged with a slide scanner using a 10× objective with 0.45 NA (Zeiss Microscopy, Axio Scan.Z1, camera: AxioCam MRm, acquisition software: ZEN Blue 2.3).

*Image processing and analysis.* Slide-scanned images were stitched using ZEN Blue 3.0 software (Zeiss Microscopy). For each section, ROIs were selected: tumor (as defined by the nuclear pattern in DAPI staining) and contralateral hemisphere avoiding the inclusion of the section margins. For image segmentation, machine learning-based software was used (ZEISS Microscopy, ZEN Intellesis). The model was trained to recognize four main classes: non-perfused vessels, underperfused vessels, perfused vessels, and background. Training annotations were done in accordance with the intravascular fluorescence signal: vessels stained only with TRITC-dextran were annotated as non-perfused, vessels stained with both TRITC-dextran and AF647-lectin were annotated as underperfused, and vessels stained only with AF647-lectin were annotated as perfused vessels. The remaining part of the image was annotated as background. The annotation principle and the performance of the trained model are illustrated on Fig. 3f, i–l. It is worth emphasizing that the training model was specifically trained to recognize the areas of dextran extravasation as a part of background so these areas were not counted as non-perfused vessels. The model performance was tested in all datasets and training iterations continued until satisfactory segmentation was achieved for most of the images. If the selection was done unsatisfactory, the basic model was refined for particular type of samples like the ones with extraordinary high level of signal from extravasated TRITC-dextran (Supplementary Fig. 11), in order to cope with the generalization problem[43]. Training dataset for the main model consisted of five large 2D images acquired with slide scanner, which were partly but extensively annotated. For modification of the model mentioned above, certain training images were replaced by the annotated images containing instances of extensive dextran extravasation providing high background levels. In images where it was not possible to achieve correct segmentation of some very high-intensity TRITC-dextran leakage spots without affecting the segmentation of other elements, regions containing these spots were deleted (Supplementary Fig. 11a, b).

As a quantitative output for the perfusion degree analysis, we used areas covered by each vascular class. Percentage of each class was calculated from total vasculature area for each section, and a weighted average was calculated for the whole tumor based on the area of each section.

Three mice bearing MC38 flank tumors (tumor volume: 152, 102, and 118 mm$^3$) and two mice bearing CT26 flank tumors (tumor volume: 106 and 93 mm$^3$) underwent experimental procedure described as protocol 3 with subsequent dissection of the tumor tissue and brain sample. Samples were further handled and imaged in the same way as GBM tumor samples.

*Additional controls.* To make sure that injected TRITC-dextran molecules are not the reason for the low transcardial perfusion degree in GBM vasculature, tumor-bearing mice underwent the same perfusion procedure as in protocol 3, but without TRITC-dextran injection and with following immunohistochemical staining of collagen IV.

### Immunohistochemistry

Coronal sections obtained from mice that underwent either standard experimental procedure for transcardial perfusion assessment as described above, or the one without TRITC-dextran injection were fixed with 4% methanol-free formaldehyde (Thermo Fisher Scientific, #28908) for 1 h. After three PBS washing steps (5 min each), the sections were blocked with a solution consisting of 5% donkey serum (Millipore, #S30-100ML) and 0.3% Triton X-100 in PBS for 3.5 h. After blocking, the samples were incubated with primary antibodies —goat anti-type IV collagen (1:500; SouthernBiotech, #1340-01) overnight at 4 °C. After three PBS washing steps, the sections were incubated with secondary antibodies—donkey anti-goat AF488 (1:500; Thermo Fisher Scientific, #A-11055) and Hoechst (Thermo Fisher Scientific, #H3570) for 4.5 h at RT. Afterward, the sections were washed and mounted with ProLong™ Diamond Antifade Mountant mounting media (Thermo Fisher Scientific, #P36970). Mounted sections were imaged with slide scanner as described above. High-magnification images were obtained using a confocal microscope using 63×, NA 1.4 objective (Zeiss Microscopy, LSM 780).

### Comparison of conventional tissue preparation versus tissue clearing for extravasation assessment

*Experimental procedure and sample preparation.* G01-bearing mice were IV-injected with TRITC-dextran, as previously described. After 2 h of TRITC-dextran circulation, mice were IV-injected with 200 µl AF647-conjugated WGA (1 µg/µl in PBS, Thermo Fisher Scientific, #W32466) to label the vasculature. After 1 h of WGA circulation, the animals were transcardially perfused with 30 ml of PBS followed by 20 ml of 4% methanol-free formaldehyde solution (Thermo Fisher Scientific, #28908). After perfusion, the brains were dissected and postfixed overnight at 4 °C. Postfixed brains were separated into two groups: one undergoing conventional sample preparation and another undergoing optical clearing. Samples from CPS group underwent cryopreservation with 15 and 30% sucrose solutions at 4 °C (steps duration: until brain's sinking). Cryopreserved samples were immersed into frozen section medium (Leica Biosystems, #3801480) and snap-frozen in liquid nitrogen-cooled isopentane. A total of 60 µm coronal sections were obtained using cryostat (Leica Biosystems, 1850 UV) and mounted in ProLong™ Diamond Antifade Mountant (Thermo Fisher Scientific, #P36970). To obtain samples for OCS group, tumors with surrounding brain tissue (to prevent damaging tumor vessels) were carefully excised. None of the sample dimensions exceeded 5 mm. The clearing technique used for OCS was based on the method developed by Klingberg et al.[65]. This approach was chosen to avoid substantial tissue delipidation, which could lead to an anisotropic sample shrinkage due to a pronounced difference in lipid content between the GBM and the normal brain tissues[66,67]. This was important when comparing OCS and sectioned samples, since the latter did not undergo delipidation. The excised samples underwent step-by-step serial dehydration in ethanol (30% for 2 h, 50% for 2 h, 70% for 2 h, 96% for 2 h, 100% overnight, and 100% for 2 h). After dehydration, the samples were incubated in ethyl cinnamate (Sigma-Aldrich, #112372) for a minimum of 6 h and were imaged shortly after the end of the last incubation step. Experimental protocols are illustrated on Fig. 4i.

*Measurements of tissue volume change due to sample processing in optically cleared versus conventionally prepared samples.* NMRI-nude mice were perfusion-fixed as described above, and the brains were postfixed overnight. After postfixation, six cylinder-shaped samples were taken using biopsy needles (diameter—3 mm). The samples were separated in the same groups and underwent the same processing steps, as described above. Before and after undergoing the preparation procedure, samples were imaged with a stereomicroscope (Zeiss Microscopy, Stemi 508; Supplementary Fig. 6a–d). The dimensions of imaged cylinders were measured in Fiji software[68] and volumes calculated to establish the correction factor (CF) for tissue volume change. CF for volume change was calculated using the following formula:

$$CF = med(OCS) - med(CPS),$$

where: med(OCS)—median value of volume change percentage from the group of optically cleared samples and med(CPS)—median value of volume change percentage from the group of conventionally prepared samples.

**Imaging, image processing, and image analysis**. Fluorescently labeled samples from both groups were imaged with a confocal laser scanning microscope using a 10× objective with 0.3 NA (Zeiss Microscopy, LSM 780, acquisition software: ZEN Black 2012 SP 5). Taking into account potential differences in signal intensity arising from difference is sample preparation methods, the microscope acquisition parameters (including Z-correction) were set to capture the intensity just below the saturation limit. Voxel size was equal to $0.69 \, \mu m \times 0.69 \, \mu m \times 3 \, \mu m$. Total imaging time of OCSs (unidirectional scan) was on average ~15 h. The size of the datasets from OCS (two channels) was on average ~60 GB. Images from CPS samples were stitched using ZEN Blue 3.0 software. Images from OCS underwent blind deconvolution in Amira 6.5 software (Thermo Fisher Scientific) and were stitched in Fiji, using Grid/Collection stitching plug-in[69]. For further quantification of dextran extravasation spots number, peripheral regions of the tumors were selected. In case of cleared samples, regions for analysis were selected from the upper quarter of the stack, characterized by the highest image quality. The volume of selected datasets was equivalent to more than $4 \, mm^3$ of postfixed tissue when corrected for tissue volume change. Numbers of dextran extravasation spots per tumor volume were independently calculated by three annotators and the resulting medians of the counts were reported.

For visualization of sectioning artifacts in the normal brain region, NMRI-nude mouse was injected with TRITC-dextran and AF647-lectin, as described above. In order to maintain dextran molecules in the healthy vasculature, animal was not perfused, but decapitated under sevoflurane anesthesia. Following steps of tissue preparation were identical to those for CPS group. High-magnification images of extravasation spots were obtained with a confocal microscope using a 63×, NA 1.4 objective (Zeiss Microscopy, LSM 780).

### Detection and analysis of extravasation in cleared tissue samples

*TRITC-dextran extravasation analysis*. Deconvolved images of optically cleared GBM samples underwent Gaussian blurring ($\sigma = 1.3$; here and below sizes of all filters are presented for the images not calibrated to the actual physical size) in ZEN Blue 3.0 and were segmented with ZEN Intellesis software (Zeiss Microscopy). The model was trained to recognize dextran extravasation spots and classify the rest of the image as background. The training dataset consisted of five different image stacks, each containing on average 13 partly annotated images. The segmentation output (3D binary masks) were then imported to Fiji and underwent postprocessing, in order to connect all parts of the segmented extravasation spot together. Post-processing consisted of four consecutive 2D dilation steps followed by 3D Gaussian filtering ($\sigma$ of $x,y,z = 3$) and thresholding (pixel intensity: 50–255). Afterward, images were imported to Amira 6.5 and extravasation spots were recognized and measured using Label analysis module. Spots smaller than $10,000 \, \mu m^3$ were excluded from the analysis. For method validation, two image subsets $2 \, mm^3$ in volume were selected from two different tumors, number of extravasation spots was counted manually by human annotator and by using developed workflow.

### Detection and analysis of albumin extravasation in cleared tissue samples.

G01 tumor-bearing mouse underwent the same experimental procedure as for dextran extravasation studies, but instead of TRITC-dextran, TRITC-conjugated albumin (50 µg/g, Thermo Fisher Scientific, #A23016) was injected. Tissue samples were optically cleared and imaged, as described above. Deconvolved images of TRITC-albumin channel in the tumor were thresholded (pixel intensity: 60–255). AF647 (lectin) channel of the same region underwent image processing steps described below (all before tracing step), and was subtracted from the thresholded TRITC-albumin channel to avoid contribution of intravascular albumin fraction to the measured volume. After subtraction, extravasated volume was measured in Fiji.

### Tracing and analysis of the vascular network in cleared tissue samples.

The AF647 channel from deconvolved images of cleared tissue samples was imported to ZEN Blue 3.0 and segmented with Zeiss ZEN Intellesis software. Models for segmenting tumor vasculature and normal brain vasculature were trained separately. Training dataset for model used for segmentation of tumor vasculature consisted of five image stacks each containing on average 18 partly annotated images. In case of the model for normal brain vasculature segmentation, four image stacks with on average three partly annotated images each comprised training dataset. Segmented 3D binary masks of the vasculature were imported to Fiji and underwent postprocessing. Postprocessing included Gaussian smoothing ($\sigma = 0.9$) followed by thresholding (pixel intensity for all tumor datasets: 170–255, and for normal brain dataset: 105–255) followed by filling the lumens of vessels segmented as hollow tubes, with subsequent median filtering (radius equal 3 for tumor datasets and 1 for normal brain dataset). For vessel lumen filling, the Fiji plug-in "Analyze Particles" was applied on Gaussian-smoothened 3D binary masks of the vessels. The plug-in was set to provide binary masks, which corresponded to the lumens of hollow vessels in a given optical section, since these were recognized as 2D particles by the plug-in. By adding these masks and the vessel masks produced by Intellesis, lumen filling was achieved in a given plane. To maximize filling of vascular lumens the procedure was done consecutively on the Z-stack sliced in XY, YZ, and XZ planes in two iterations. Sequence of operations for achieving lumen filling is depicted in detail in Supplementary Fig. 7a. Postprocessed datasets were imported to Amira 6.5 and the vasculature was traced by the AutoSkeleton module employing method

developed by Fouard et al.[70]. Vascular geometry was calculated in Euclidean space through calculation of the Euclidean distance map, and vascular segments shorter than 10 µm were excluded from analysis. An underestimation of vessel diameter produced by the method was measured on 100 vessel segments with different diameter. These measurements were further used for the curve fitting with exponential decay function, which allowed to calculate the CF with the following formula:

$$CF = (Y0 - Plateau) \times \exp(-K \times MD) + Plateau$$

where: $Y0 = 3.7$, Plateau $= 1.3$, $K = 0.5861$, MD—initially measured diameter of a vessel segment.

For correcting initial diameter measurements, they were multiplied by the value of the CF.

Of note, in the vasculature spatial graphs shown in the figures and Supplementary Movie 2, were not scaled 1:1 with regard to the vessel diameter, in order to provide a better overview on the obtained spatial graph.

### Software for machine learning-based segmentation and workstation characteristics.

The tool used for image segmentation in the present study is the Python-based deep learning software package ZEN Intellesis (Zeiss Microscopy)[71]. The software allows for a supervised training of the pixel classifier on the microscopy images. For feature extraction the software employs very deep convolutional neural network the VGG-19, which was adopted by Zeiss Microscopy from the work of Simonian and Zisserman[72] without any considerable changes or additional training. According to the authors, the training of the neural network was done using mini-batch gradient descent as optimization algorithm for multimodal logistic regression objective, the batch size was 256 and the number of epochs was 74, for more details please refer to the original work[72]. From the third layer of the VGG-19, 256 features were extracted and used to train random forest classification algorithm[73]. For the structured predictions, conditional random fields[74] were applied for all datasets except normal brain vasculature segmentation of which was satisfactory with basic settings.

The workstation used for image analysis had following characteristics, CPU: dual Xeon processors E5-2670, 2.6 GHz (96 GB RAM) and GPU: GeForce RTX 2080Ti (11 GB RAM).

### Statistics and reproducibility.

Statistical analysis was mainly performed using GraphPad Prism 8.1.1 (GraphPad software). To estimate the difference between two unpaired groups, the two-tailed Mann–Whitney $U$ test was used. For the comparison of paired groups, the two-tailed Wilcoxon signed-rank test was used. To compare the proportions of different vascular classes, we initially arcsine converted the percentages and performed logistic regression to assess changes (done using R for Windows, version 2.12.1). A $P$ value $<0.05$ was considered statistically significant, and the degree of significance is represented as follows: $*P < 0.05$, $**P < 0.01$, $***P < 0.001$, and $****P < 0.0001$. Sample sizes are specified in the figure legends containing graphs where statistical information is derived. All data represented in the main figures as graphs can be found in the form of tables in Supplementary Data 1.

**Reporting summary**. Further information on research design is available in the Nature Research Reporting Summary linked to this article.

## Data availability

Data underlying graphs presented in the main figures are available in the form of excel sheets as Supplementary Data to the present paper. Other data are available upon reasonable request by contacting the corresponding authors.

## Code availability

Code for image postprocessing is available upon request by contacting the corresponding authors.

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

## Acknowledgements
Microscopy imaging and image analysis described in the present paper was done using the instruments at the Core Facility for Integrated Microscopy, Department of Biomedical Sciences, University of Copenhagen. We are thankful to Esben Christensen and Trine Bjørnbo Engel, Department of Health Technology, Technical University of Denmark, for providing syngeneic colorectal cancer tumor models. We thank Clara Prats Gavalda and Pablo Varas, Core Facility for Integrated Microscopy, Department of Biomedical Sciences, University of Copenhagen and Sebastian Rhode from Zeiss Microscopy for sharing their expertise. We are thankful to Nanna Elmstedt Bild, Department of Health Technology, Technical University of Denmark for assisting with creation of schematic drawings for the present paper. This research was funded by generous grants from the Lundbeck Foundation Research Initiative on Brain Barriers and Drug Delivery (Grant no. R155-2013-14113).

## Author contributions
S.K., K.B.J., C.H. and T.L.A. designed the study with an input from T.H.B., P.H., and A.E.H. S.K., K.B.J., T.H.B., J.M.G., F.P.F., E.A.A.O. and A.E.H. performed the experimental work. S.K. developed the workflows and analyzed the data in collaboration with K.B.J., T.H.B. and C.H. S.K., K.B.J., T.H.B., J.M.G., F.P.F., E.A.A.O., P.H., A.E.H., A.K., C.H. and T.L.A. took part in data interpretation. S.K., K.B.J. and C.H. drafted the manuscript, which was finalized together with T.L.A. T.H.B., J.M.G., F.P.F., E.A.A.O., P.H., A.E.H., and A.K. took part in editing the manuscript.

## Competing interests
The authors declare no competing interests.
