## [Peer Review File · Communications Biology]

Reviewers' comments:

Reviewer #1 (Remarks to the Author):

Although the use of optical clearing methods to assess extravasation in tumors is not novel (Lee SS, Bindokas VP, Kron SJ. Multiplex Three-Dimensional Mapping of Macromolecular Drug Distribution in the Tumor Microenvironment. *Mol Cancer Ther.* 2019 Jan;18(1):213-226. doi: 10.1158/1535-7163.MCT-18-0554. Epub 2018 Oct 15. PMID: 30322947; PMCID: PMC6318001.), the authors present useful warnings to obtain more accurate results in drug delivery studies. This paper follows a robust experimental approach, with no apparent flaws in its methodology. It also provides deep insight into image processing techniques that are good candidates to solve, in a generalizable way, common problems that arise when segmenting vasculature and other elements of 3D microscopy images. Authors present Machine Learning in a justified way as a valuable tool to solve most technical issues concerning data analysis in drug delivery systems.

1. One of the main claims of the paper is the demonstration of the current flaws of extravasation studies of tumoral tissue with quantitative assays, demonstrating the superiority of microscopy techniques in these types of studies. This claim would be more convincing if the authors mentioned references demonstrating that intermittent vascular perfusion is not currently considered as a factor hindering drug delivery studies (line 60, line 303)

2. Although optical clearing methods are a nice choice to optimize extravasation studies, clearing techniques often use aggressive compounds that can affect the tissue's structure and properties, and not only its size (line 481). In fact, some optical clearing methods may actually increase the permeability of the BBB. Therefore, the results of extravasation spots count in OCS may not be entirely accurate. This should be indicated somewhere in the text.

Minor comments:

1. Reference 20 of line 84 does not seem to prove the importance of transcardial perfusion in this type of studies.

2. On page 21, it would be nice to add some reference or derivation of the CF formula.

3. There is a missing definition of scale bars in Figure 5H, Supplementary Figure 2 D and E, Supplementary Figure 6 E and F.

4. Regarding the vessel lumen filling strategy presented on page 20, a further explanation of the "analyze particles" implementation to obtain binary masks of the lumen would be of great help to facilitate reproducibility of the workflow.

Reviewer #2 (Remarks to the Author):

In the manuscript titled "Optical tissue clearing and machine learning can precisely characterize extravasation and blood vessel architecture in brain tumors", the authors discuss the problem of the quantification of drug accumulation in brain tissue.

The authors present the difficulties that arise in the assessment of drug delivery in brain tumor tissue. First of all, they show that transcardial perfusion -- which is essential in removing the circulating drug from vessels -- is hindered in samples of glioblastoma (GBM) because of the pathologically-induced changes in the vasculature morphology. This problem alone is likely to cause a wrong assessment of compound extravasation. Continuing their line of reasoning, they also show that conventionally prepared samples undergoing cryosectioning are prone to overestimation of the amount of extravasated drug because of the actual damage inflicted to the vessels during the cutting procedure.

With these two findings, the authors show that 3D imaging of optically cleared tissue -- leaving the sample intact -- is an effective method for simultaneous precise assessment of drug extravasation and angioarchitecture reconstruction. This also allows them to link the degree of extravasation to specific features of the brain tumor vasculature.

The manuscript is well written and easy to follow. The techniques used throughout this work are not new if taken independently. The main novelty of this work is the combination of several techniques (tissue clearing, 3D deep imaging, vessel segmentation using machine learning approaches) to tackle the problem of precise extravasation assessment, and the demonstration that transcatheter perfusion combined with tissue sectioning is not an effective method for this aim, both of which could be of interest for the scientific community. For machine learning, the authors use commercially available solutions (i.e. no actual model development was done in this work), which they consider to be an advantage. Overall, the work appears to be meticulous and sound from a procedural point of view.

Here follows a list of minor points and suggestions that the authors might find useful:

- * Figure 3M-P: it might help the reader if the authors added a small label within the panels indicating to which cell lines the histograms refer to
- * Line 475: "liquid nitrogen-collid" is probably a typo for "liquid nitrogen-cooled".
- * Figure 6D: it might help to add a scale to the color bar, or at least indicate min/max values
- * Supplementary Figure 2, D-F: the label of the vertical scale is missing, probably it's the same as panel G
- * It might be of interest to know a few more details on their confocal imaging setup (e.g. voxel size, total acquisition time, dataset size, etc).

Reviewer #3 (Remarks to the Author):

In the manuscript entitled "Optical tissue clearing and machine learning can precisely characterize extravasation and blood vessel architecture in brain tumors", Serhii Kostrikov et al. used optical tissue clearing methods and machine learning to quantify compound extravasation models of brain tumors and reconstruct blood vessel structures. The comments are given as follows:

1. In the manuscript, optical clearing was used to avoid the level of overestimation arising from sectioning artifacts. However, almost all optical cleaning methods will cause fluorescence loss, which will affect the subsequent statistical results. This article does not consider how this factor will affect the experimental results;
2. In Fig. 2A, lectin is used for cardiac perfusion with PBS, which is performed after dextran. But in Fig. 4I, lectin and dextran are injected and circulated together. Is there any difference between these two experimental operations?
3. There are some problems with the statistical results in the article. For example, the error bar in Fig.5H is too large, some results in Fig. 7 do not have error bar. These statistics are not credible.
4. Please specify the hyperparameters (such as learning rate, batch size, epoch, etc) and optimizer used in the machine learning model.
5. Please specify the dataset used when training the machine learning model.
6. When performing segmentation, why not use the commonly used U-Net, but use VGG19 to extract 256 features and classified by random forest? Please compare and provide the advantages of this method.
7. It is better to list the specific information of CPU, GPU and RAM when describing machine learning method.
8. Please pay attention to keep the consistency of each image, such as text size, etc.

Reviewer's comment	Response
Reviewer #1	We are very grateful to the reviewer for investing the time to examine our work in-depth and providing such positive and valuable feedback for improving the manuscript.
1. One of the main claims of the paper is the demonstration of the current flaws of extravasation studies of tumoral tissue with quantitative assays, demonstrating the superiority of microscopy techniques in these types of studies. This claim would be more convincing if the authors mentioned references demonstrating that intermittent vascular perfusion is not currently considered as a factor hindering drug delivery studies (line 60, line 303)	We fully agree that adding references to the mentioned statements makes them more convincing and thank the reviewer for pointing this out. We have slightly extended the sentence in lines 303-304 and added a number of references to the works employing transcordial perfusion for studying compound extravasation in tumor tissue assuming efficiency of transcordial perfusion in blood removal: “Even though the phenomenon of intermittent vascular perfusion in GBM has been known for years³⁴, it has not been considered as a factor causing serious methodological limitations in drug delivery studies, and transcordial perfusion has been widely employed for blood removal in tumor vessels^{19,21,35-37}.”
2. Although optical clearing methods are a nice choice to optimize extravasation studies. clearing techniques often use aggressive compounds that can affect the tissue's structure and properties, and not only its size (line 481). In fact, some optical clearing methods may actually increase the permeability of the BBB. Therefore, the results of extravasation spots count in OCS may not be entirely accurate. This should be indicated somewhere in the text.	We thank the reviewer for rising this important point. Preserving an adequate picture of BBB permeability and tissue structure in general was one of our main concerns. Therefore, the samples were perfusion-fixed and postfixed with methanol-free (as an additional precaution for minimizing the effects on BBB permeability) formalin for 24 hours prior to starting the clearing procedure. To prevent any possible fixation deterioration over time, we proceeded with gradual dehydration steps straight away after completing the postfixation with only brief washing in PBS. Since the dehydration also contributes to the sample fixation, it is safe to assume that the distortion of the extravasation picture due to presumed changes in BBB permeability was negligible. In general, the optical tissue clearing protocol that we have chosen is fast and have comparatively mild effects on the tissue. In addition, it did not include dichloromethane incubation step (an aggressive delipidation agent). The imaging was done during the first day after clearing procedure completion, in order to avoid any potential tissue changes resulting from long-term incubation in refractive index-matching agent ethyl cinnamate. At the same time, we agree that even with all listed considerations taken into account, current literature lack information about every potential effect of complete dehydration and RI-matching with ethyl cinnamate on the tissue and extravasation patterns in particular, which are yet to be studied. Therefore, we agree that there can be a possibility of confounding extravasation assessment with such sample processing. A few lines indicating such risks were added (lines 322-327): “It is also important to note that various tissue clearing methods often use aggressive compounds⁴⁰, which can have versatile effects on the tissue as well as extravasation patterns, and are yet to be comprehensively studied. Such potential effects can be limited by using optical clearing methods with relatively mild tissue treatment like the one described in the present study, on one hand, and, on the other, by ensuring robust chemical fixation of the studied compound and the tissue sample in general.”
3. Reference 20 of line 84 does not seem to prove the importance of transcordial perfusion in this type of studies.	We found the mentioned work to be relevant for referencing because the authors used transcordial perfusion for their vessel permeability assay, which was based on analyzing Evans Blue extravasation. As described in the subsection “vessel permeability assay” in Materials and Methods section, in their experiments, transcordial perfusion was used for removing the blood and previously injected Evans Blue from the tumor vasculature. This was followed by incubation in formamide for the dye extraction and subsequent absorbance measurements. The transcordial perfusion was used for presumed removal of the intravascular fraction of the Evans Blue in order to obtain only extravascular fraction of the compound in the solution for absorbance measurements.

4. On page 21, it would be nice to add some reference or derivation of the CF formula.	The extended explanation is now added in lines 686-695: “An underestimation of vessel diameter produced by the method was measured on 100 vessel segments with different diameter. These measurements were further used for the curve fitting with exponential decay function, which allowed to calculate the correction factor (CF) with the following formula. For correcting initial diameter measurements, they were multiplied by the value of the CF.”
5. There is a missing definition of scale bars in Figure 5H, Supplementary Figure 2 D and E, Supplementary Figure 6 E and F.	We are thankful that these errors were pointed out, they are now corrected.
6. Regarding the vessel lumen filling strategy presented on page 20, a further explanation of the “analyze particles” implementation to obtain binary masks of the lumen would be of great help to facilitate reproducibility of the workflow.	We thank the reviewer for drawing additional attention to the reproducibility of the workflow. In order to facilitate the communication of the workflow details in the best possible way, we slightly extended an explanation in the method section (lines 678-679) and added a schematic depiction of the workflow stages as well as its effects on the image. This is now depicted in the newly added supplementary figure (Supplementary fig. 7), which is referred to in lines 237, 238, 407 and 683. In addition, we are ready to provide the code for postprocessing of the binary masks of vasculature (which include vessel lumen filling) upon request, and we added this statement in lines 735-737. Lines 678-679: “The plug-in was set to provide binary masks, which corresponded to the lumens of hollow vessels in a given optical section, since those were recognized as 2D particles by the plug-in.” Lines 735-737: “Code availability: Code for image postprocessing is available upon request by contacting the corresponding authors.” Newly added supplementary figure 7 with figure legend:  Supplementary Figure 7. Performance of the in-house developed workflow for the adjustable vessel lumen filling. (a) Schematic depiction of one iteration of the developed workflow for the adjustable filling of the vessel lumens. (b) Binary mask of the vessels after filling holes operation in the xy plane demonstrating the lumen above established size limit not being filled. (c) Binary mask of the vessels after filling holes in xz plane (cyan) and yz plane (blue). (d) Binary mask of the vessels after the first iteration of the lumen filling workflow and 2D median filtering demonstrating the lumen above established size limit being filled. Scale bars, 50µm. In b-d an arrow point at the large vessel segmented as a hollow tube. 3D rendering of the vasculature (white) with lumen filled using standard “fill hole” operation in Fiji (magenta) (e), and using developed workflow (cyan) (f), an arrow points on the space between small vessel being erroneously filled. Binary mask of the vessel

	lumens produced by standard “fill hole” operation in Fiji (g), and developed workflow (h), arrows point on the imprecisions in the mask produced by standard “fill hole” operation. Binary masks of the vessels after lumen filling done using standard “fill hole” operation in Fiji (i) and developed workflow (j) arrows point on the spaces surrounded by small capillaries being filled by standard “fill hole” operation in Fiji. Scale bars, 100µm.
Reviewer #2	We are very grateful to the reviewer for such thorough examination of our work and positive feedback arising from comprehensive understanding of our work with regard to both methodological details as well as general context.
1. Figure 3M-P: it might help the reader if the authors added a small label within the panels indicating to which cell lines the histograms refer to	The figure is now modified in accordance to the comment:  Figure 3M-P: A multi-panel figure showing fluorescence microscopy images and histograms. Panels a-d show TRITC-dextran (green), AF647-lectin (red), and Collagen IV (cyan) staining. Panels e-h show processed images and histograms of pixel intensity for underperfused (blue) and non-perfused (green) vessels. Panels i-l show binary masks and segmentation. Panels m-p show stacked bar charts of % from total vasculature for cell lines G01, G06, U87, and All combined, comparing Contralateral hemisphere and Tumor regions.
2. Line 475: "liquid nitrogen-colled" is probably a typo for "liquid nitrogen-cooled".	We are thankful for spotting this typo, which is now corrected.
3. Figure 6D: it might help to add a scale to the color bar, or at least indicate min/max values	The scale indicating min/max values is now added to all figures containing spatial graphs with such color-coding, which are Fig. 6, Fig. 7 and Supplementary Fig. 10. In all mentioned figures, the scale was added in the way exemplified below (excerpt from Fig. 6):

Following addition to the figure legends was made after color-coding specification: “numbers for minimum and maximum vessel diameter are presented in μm .”

4. Supplementary Figure 2, D-F: the label of the vertical scale is missing, probably it's the same as panel G

The vertical scale is placed properly now, so it serves all the panels:

5. It might be of interest to know a few more details on their confocal imaging setup (e.g. voxel size, total acquisition time, dataset size, etc).

The information is now added to the method section (lines 627-628):

“Voxel size was equal to $0.69 \mu\text{m} \times 0.69 \mu\text{m} \times 3 \mu\text{m}$. Total imaging time of OCSs (unidirectional scan) was on average ~ 15 hours. The size of the datasets from OCS (two channels) was on average ~ 60 GB.”

Reviewer #3

We thank the reviewer for investing the time to consider our work, providing valuable comments for the manuscript improvement and rising many interesting and important points of discussion.

1. In the manuscript, optical clearing was used to avoid the level of overestimation arising from sectioning artifacts. However, almost all optical cleaning methods will cause fluorescence loss, which will affect the subsequent statistical results. This article does not consider how this factor will affect the experimental results;

We are thankful to the reviewer for rising such an important concern. Many clearing protocols can indeed be deteriorating to the fluorophores in the tissue. However, this is largely a concern for endogenous fluorescent proteins (e.g. GFP or YFP), which are sensitive to organic solvent-based clearing procedures and have a half-life in final clearing solution of up to two days¹. In our study, we used TRITC-labelled dextran for extravasation analysis. TRITC is characterized by a much higher stability compared to the fluorescent proteins. Furthermore, its stability is comparable with Alexa Fluor dyes, which have been reported to be stable for many months in index-matching solutions². In addition, we would also like to emphasize that by the time of imaging, the samples were incubated in refractive index-matching solution for 6 hours only. Furthermore, for manual extravasation spot quantification, we used the dataset from an upper quarter of the tumor, which has been imaged the first. Taking all these factors into account, we assume the fluorescent loss in the studied samples to be negligible.

1. Ertürk, A. *et al.* Three-dimensional imaging of solvent-cleared organs using 3DISCO. *Nat. Protoc.* **7**, 1983–1995 (2012).
2. Renier, N. *et al.* IDISCO: A simple, rapid method to immunolabel large tissue samples for volume imaging. *Cell* **159**, 896–910 (2014).

2. In Fig. 2A, lectin is used for cardiac perfusion with PBS, which is performed

A fundamental difference between the two experimental operations is that in case of introducing the lectin through transcordial perfusion, it can only reach the vessels in the tumor, which are accessible for the circulation during the few minutes of the procedure.

after dextran. But in Fig. 4I, lectin and dextran are injected and circulated together. Is there any difference between these two experimental operations?	In such settings, lectin circulation can be affected by a decrease in the vessel tone of the dying animal. Contrary, when the lectins are introduced one hour prior to the endpoint through intravenous injection, they can provide a much more extensive labelling of the tumor vasculature, since they can reach the vessels with perfusion cycles below one hour and are delivered through physiological circulation with proper vessel tone maintained. The fact that dextran and lectin share one hour of circulation (after two hours of dextran circulation only) is not presumed to affect the outcome of the experiment, since no interactions between the two molecules are expected (the specific binding site of the WGA is N-Acetyl-D-glucosamine³, which is not a part of dextran chemical structure⁴). 3. Gallagher, J. T. Carbohydrate-binding properties of lectins: A possible approach to lectin nomenclature and classification. Biosci. Rep. 621–632 (1984). 4. Neely, W. B. Dextran: Structure and Synthesis. Adv. Carbohydr. Chem. 15, 341–369 (1961).
3. There are some problems with the statistical results in the article. For example, the error bar in Fig.5H is too large, some results in Fig. 7 do not have error bar. These statistics are not credible.	The error bars in the fig. 5h is large because they reflect standard deviation of extravasation spot counts in the validation datasets, which were taken from two different tumors and different regions with regard to imaging depths. Generally, it is natural to have a highly heterogeneous extravasation patterns both between the tumors as well as within one tumor^{5,6}. Therefore, in this context, we consider large error bars to be expected and not problematic for any of the study conclusions. In Fig. 7h,m,n the error bars are not shown because the data represents single tissue subset derived from one single tumor to exemplify how interregional differences can be determined with our workflow. 5. Jain, R. K. & Stylianopoulos, T. Delivering nanomedicine to solid tumors. Nature Reviews Clinical Oncology (2010). doi:10.1038/nrclinonc.2010.139 6. Sarkaria, J. N. et al. Is the blood-brain barrier really disrupted in all glioblastomas? A critical assessment of existing clinical data. Neuro. Oncol. 20, 184–191 (2018).
4. Please specify the hyperparameters (such as learning rate, batch size, epoch, etc) and optimizer used in the machine learning model.	This information is now added to the method section (lines 703-708). We have also added a note that the network was directly adopted from the work of Simonian and Zisserman⁷ without any additional training, and that all methodological details can be found in the original work. Please see the excerpt below: “For feature extraction the software employs very deep convolutional neural network the VGG-19, which was adopted by Zeiss Microscopy from the work of Simonian and Zisserman⁷³ without any considerable changes or additional training. According to the authors, the training of the neural network was done using mini-batch gradient descent as optimization algorithm for multimodal logistic regression objective, the batch size was 256 and the number of epochs was 74, for more details please refer to the original work⁷³. From the 3rd layer of the VGG-19, 256 features were extracted and used to train random forest classification algorithm⁷⁴” 7. Simonyan, K. & Zisserman, A. Very Deep Convolutional Networks for Large-Scale Image Recognition. Proc. Int. Conf. Learn. Represent. 1–14 (2014).
5. Please specify the dataset used when training the machine learning model.	We have added the information regarding the training datasets used in the present study, as well as training procedure in general. Please see lines 203 and 648-649 for the model used to segment dextran extravasation in 3D images from cleared tissue samples, lines 552-555 for models used to segment intravascular dextran and lectin stainings, for perfusion degree measurements in 2D images from slide scanner, and lines 669-672 for models used for vessel segmentation in 3D images from cleared tissue samples. Lines 202-203: “For segmentation of extravasation spots, the model was trained on image stacks from different tumor regions and different imaging depths.” Lines 552-555: “Training dataset for the main model consisted of five large 2D images acquired with slide scanner, which were partly but extensively annotated. For modification of the model mentioned above, certain training images were replaced by the annotated images containing instances of extensive dextran extravasation providing high background levels.”

	Lines 648-649: “The training dataset consisted of five different image stacks, each containing on average 13 partly annotated images.” Lines 669-672: “Training dataset for model used for segmentation of tumor vasculature consisted of five image stacks each containing on average 18 partly annotated images. In case of the model for normal brain vasculature segmentation, four image stacks with on average three partly annotated images each comprised training dataset.”
6. When performing segmentation, why not use the commonly used U-Net, but use VGG19 to extract 256 features and classified by random forest? Please compare and provide the advantages of this method.	We are thankful for this interesting question, as it is definitely important to discuss. One of our main priorities in designing the image analysis workflows was to make them user-friendly and possible to employ and customize for biomedical scientists without a specific background in programming and machine learning. The reasons behind our choice favoring approach like Intellesis were two-fold. On one hand, Intellesis require much smaller amounts of training data (~10-100 annotated samples), and hence much less hands-on work compared to training a neural network from scratch, which would require thousands of annotated samples in a training dataset⁸. This is due to the fact that while Intellesis takes advantage of deep learning, it employs the neural network with pre-trained weights only for feature extraction, which are then passed on to random forest classifier and used for pixel classification. In such settings, no training is required for the network itself, but for the classifier only. Another point of consideration for us was availability of a user interface in the segmentation software, because training a neural network requires a certain expertise in programing and deep learning. When we were designing the presented workflows, there were no software or plug-ins providing a user interface for training U-Net (to our knowledge, he first manuscript describing such development was deposited 16th of October 2019 on bioRxiv⁹). We would like to emphasize that while we consider machine learning and deep learning in particular to be optimal approaches for segmentation of heterogeneous images from optically cleared samples and definitely key parts of the developed workflows, the development of a deep learning-based segmentation method itself was not our purpose. Thus, meticulous comparison of different machine learning-based segmentation approaches lies outside of the scope of our manuscript. Therefore, we do not discourage people to try other machine learning-based approaches as a replacement for Intellesis in the presented workflow including U-Net. A part of discussion section touching upon the workflows was slightly extended with addition of brief comparison between Intellesis and neural networks in general (lines 335-341): “It is also important to note that while the exemplified segmentation approach takes advantage of deep learning (through employing a convolutional neural network with pre-trained weights for feature extraction, which are used afterwards for pixel classification by random forest of decision trees), it is not as time-consuming as training of a neural network from scratch⁴³. This is because the used neural network (VGG-19) is already pre-trained and training is required for the classifier only, which needs orders of magnitude less training data compared to a neural network.” 8. Belthangady, C. & Royer, L. A. Applications, promises, and pitfalls of deep learning for fluorescence image reconstruction. Nat. Methods (2019). doi:10.1038/s41592-019-0458-z 9. Gómez-de-Mariscal, E. et al. DeepImagej: A user-friendly plugin to run deep learning models in ImageJ. Bioarxiv (2019).
7. It is better to list the specific information of CPU, GPU and RAM when describing machine learning method.	We thank the reviewer for pointing this out, the requested information is now added to the method section (lines 711-712): “The workstation used for image analysis had following characteristics, CPU: dual Xeon processors E5-2670, 2.6 GHz (96 GB RAM), GPU: GeForce RTX 2080Ti (11 GB RAM).”
8. Please pay attention to keep the consistency of each image, such as text size, etc.	We are thankful for drawing our attention to this issue, all the figures are now examined and the inconsistencies are corrected.

REVIEWERS' COMMENTS:

Reviewer #2 (Remarks to the Author):

Compared to the first submission, the authors have satisfactorily addressed the issues that were raised by the reviewers. The manuscript can now be considered for publication.

Reviewer #3 (Remarks to the Author):

The author's response in general is satisfactory. I have no other questions and recommend publishing.

We are very delighted to see such positive referees' comments. We are thankful to the reviewers for investing their time and expertise into considering our manuscript, as well as for the fair and efficient review process.